# Single-cell transcriptomics identifies altered neutrophil dynamics and accentuated T-cell cytotoxicity in tobacco-flavored e-cigarette-exposed mouse lungs

**Gagandeep Kaur, Thomas Lamb, Ariel Tjitropranoto, Irfan Rahman***

Department of Environmental Medicine, University of Rochester Medical Center, Rochester, United States

**\*For correspondence:**
Irfan_Rahman@urmc.rochester.
edu

**Competing interest:** The authors declare that no competing interests exist.

## eLife Assessment

This manuscript by Kaur et al. identifies differential gene expression in distinct cell populations, specifically myeloid and lymphoid cells, following short-term exposure to e-cigarette aerosols with various flavors. Their findings are **useful** because they provide a single-cell sequencing data resource for assessing which genes and cellular pathways could be affected by e-cig aerosols and their components. However, the evidence is **incomplete** due to limited number of biological replicates per condition, as well as due to the lack of in vivo validation.

**Abstract** Despite the growing public health threat of electronic cigarettes (e-cigs), the cell-specific immune responses to differently flavored e-cig exposure remain poorly understood. To bridge this gap, we characterized the lung immune landscape following acute nose-only exposure to flavored e-cig aerosols in vivo using single-cell RNA sequencing (scRNA seq) in mice. Metal analysis of daily generated aerosols revealed flavor-dependent, day-to-day variation in metal (Ni, Cu, K, and Zn) leaching. scRNA seq profiling of 71,725 lung cells from control and exposed mice revealed pronounced dysregulation of myeloid cell function in menthol (324 differentially expressed genes, DEGs) and tobacco (553 DEGs) flavors, and lymphoid cell dysregulation in fruit-flavor (112 DEGs) e-cig aerosol exposed mouse lung, compared to air controls. Flow cytometry corroborated these findings, showing increased neutrophil frequencies and reduced eosinophil counts in menthol- and tobacco-exposed lungs. Flavored e-cig exposure also increased CD8+ T-cell proportions, upregulated inflammatory gene expression (*Stat4*, *Il1b*, *Il1bos*, *Il1ra*, and *Cxcl3*), and enriched terms like 'Th1 cytokine signaling' and 'NK cell degranulation'. Notably, tobacco-flavored e-cig aerosol exposure increased immature (Ly6G−) neutrophils and reduced S100A8 expression, suggesting altered neutrophil activation in vivo. Overall, this study identifies flavor-dependent immune alterations in the lung following acute e-cig aerosol exposure and provides a foundation for future mechanistic studies.

## Introduction

Electronic cigarettes (e-cigs) or electronic nicotine delivery systems are a relatively novel set of tobacco/nicotine and flavored products that have gained immense popularity among adolescents and young adults in many countries including the United States (US), the United Kingdom (UK), and China. Flavors are one of the key features that make these products alluring to the younger diaspora (*Ma et al., 2022*). Reports indicate that in 2020 about 22.5% of high school students and 9.4% of middle

school students in the US were daily vapers or e-cig users with fruit (66%), mint (57.5%), and menthol (44.5%) being the most commonly used flavors (*Wang et al., 2020a*). However, not much is known about the flavor-specific effects of e-cig vaping on the health and immunity of an individual, especially focusing on cell types and gene expressions.

E-cig products and aerosols are known to contain harmful constituents including formaldehyde, benzaldehyde, acrolein, *n*-nitrosamines, volatile organic compounds, ketenes, and metal ions (*Wu and O'Shea, 2020*; *Goniewicz et al., 2013*; *Lee et al., 2020*). Studies have indicated that exposure to e-cigs may enhance inflammatory responses, oxidative stress, and genomic instability in exposed cells or animal systems (*Wang et al., 2020b*; *Muthumalage et al., 2019*; *Lee et al., 2018b*). Risk assessment (systemic) of inhaled diacetyl, a potential component of e-liquids, has estimated the non-carcinogenic hazard quotient to be greater than 1 among teens (*White et al., 2021*). Furthermore, clinical and in vivo studies have suggested that exposure to e-cig aerosols could impair innate immune responses in the host, thus making them more susceptible to bacterial/viral infections. The bacterial clearance, mucous production, and phagocytic responses in these individuals are shown to be affected upon use of e-cigs (*Martin et al., 2016*; *Sussan et al., 2015*; *Masso-Silva et al., 2021*; *Madison et al., 2019*; *Cao et al., 2021*).

However, cell-specific changes within the lung upon vaping are not fully understood, making it hard to determine the health impacts of the use of these novel products. In this respect, single-cell technology is a powerful tool to analyze gene expression changes within cell populations to study cellular heterogeneity and function (*Jovic et al., 2022*; *Inayatullah et al., 2025*; *Ke et al., 2022*). Such an investigation is important to deduce the health effects of acute and chronic use of e-cigs in young adults. In this study, we aim to determine the effects of acute exposure to e-cig aerosols on mouse lungs at single-cell level. To do so, we exposed C57BL/6J mice to 5-day nose-only exposure to air, propylene glycol:vegetable glycerin (PG:VG), fruit-, menthol-, and tobacco-flavored e-cig aerosols. The nose-only exposure has more translational relevance over the whole-body exposure (*Kogel et al., 2021*), owing to which, we chose nose-only exposure profile for this work. To limit the stress to the animals, a 1-hr exposure was chosen per day. We performed single-cell RNA sequencing (scRNA seq) on the lung digests from exposed and control animals and identified neutrophils and T cells, among others, as the major cell populations in the lung that were affected upon acute exposure. We were able to identify 29 gene targets that were commonly dysregulated among all our treatment groups upon aggregating results from the major lung cell types. These gene targets are the markers of early immune dysfunction upon e-cig aerosol exposure in vivo and could be studied in detail to understand temporal changes in their expression and function that may govern allergic responses and adverse pulmonary health outcomes upon acute and sub-acute exposures to e-cigarette aerosols.

## Results

### Exposure to e-cig aerosols results in flavor-dependent exposure to different metals and mild histological changes in vivo

This study was designed to characterize the effects of exposure to flavored e-cig aerosols at single-cell level to understand the immunological changes in the lung microenvironment. To do so, we generated a single-cell profile of e-cig aerosol exposed mouse lungs (*n* = 2/sex/group). The thus obtained results were then validated with the help of our validation cohort of *n* = 3/sex/group as shown in *Figure 1A*. Since all the commercially available e-liquids used in this study contained tobacco-derived nicotine (TDN), we first determined the levels of serum cotinine (a metabolite of nicotine) to prove successful exposure of the mice in each treatment group. As expected, we did not see any traces of cotinine in the serum of air and PG:VG exposed mice. Significant levels of cotinine were detected in the serum of mice exposed to fruit-, menthol-, and tobacco-flavored e-cig aerosols (*Figure 1—figure supplement 1B*), validating successful exposure of our test animals to TDN containing flavored e-cigs.

Since metals released upon heating of the coils of e-cig devices are a source of toxicity upon vaping (*Olmedo et al., 2018*; *Aherrera et al., 2023*), we further monitored the levels of metals in the e-cig aerosols generated during each day of mouse exposures. This acted as an indirect measure for characterizing the chemical properties of the aerosols used for exposure in this study. To monitor the release of metals into the mouse lungs, the aerosol condensate from each day of exposure was collected and the levels of select elements were detected using inductively coupled plasma mass

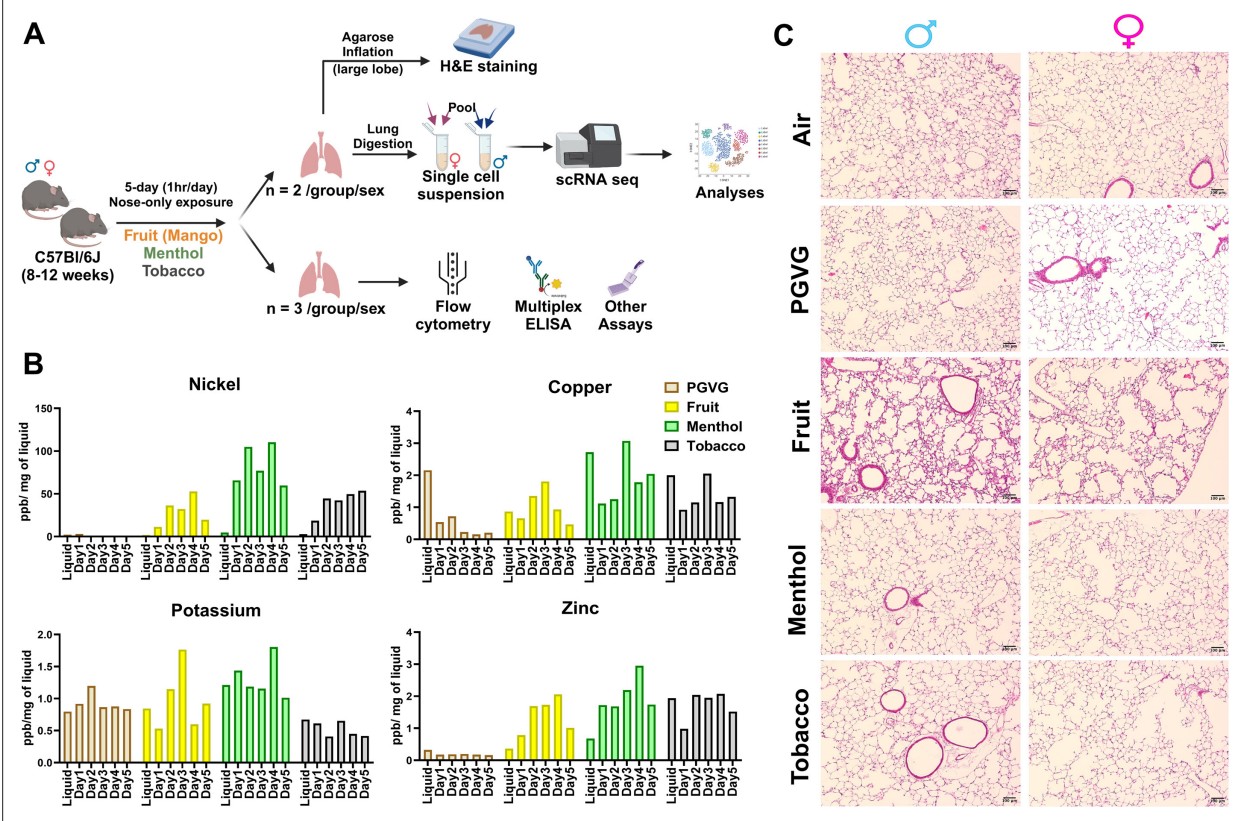

**Figure 1.** Flavor-dependent changes in the levels of quantified metals, but no major histological damage on acute exposure to flavored e-cig aerosol in C57BL/6J mice. Schematics showing the exposure profile and experimental design to understand the effects of exposure to differently flavored (fruit, menthol, and tobacco) e-cig aerosols in the lungs of C57BL/6J mice using scRNA seq (**A**). Bar graph showing the levels of metals (*Figure 1—source data 1*) as determined by inductively coupled plasma mass spectrometry (ICP-MS) in the aerosols captured daily during exposures using inExpose nose-only inhalation system from Scireq technologies (**B**). Lung morphometric changes observed using hematoxylin and eosin (H&E) staining of lung slices from air, PG:VG, and differently flavored e-cig aerosol exposed mice lungs. Representative images of *n* = 2/sex/group at ×10 magnification are provided (**C**).

The online version of this article includes the following source data and figure supplement(s) for figure 1:

**Source data 1.** Day-wise levels of Ni, Cu, K, and Zn as determined by inductively coupled plasma mass spectrometry (ICP-MS) in the aerosols captured daily during exposures using the inExpose nose-only inhalation system from Scireq technologies as plotted in *Figure 1B*.

**Figure supplement 1.** Schematics and characteristics of exposure system used for in vivo experiments.

**Figure supplement 1—source data 1.** Serum cotinine levels in the blood of differently flavored e-cig aerosol exposed and control (air and PG:VG) C57BL/6J mice as plotted in *Figure 1—figure supplement 1B*.

spectrometry (ICP-MS). A detailed account of the concentrations of identified elements/metals is provided in *Table 1*. Interestingly, we identified flavor-dependent changes in the levels of metals like Ni, Zn, Na, K, and Cu on a day-to-day basis. Note, despite the use of the same wattage and temperature (max of 230°C) for generation of e-cig aerosols, the leaching of each metal varied per day of exposure (*Figure 1B*). This is a crucial result as it highlights the importance of studying the impact of atomizer, coil composition, and design on the chemical composition of the generated aerosols. These variations might affect the risk and toxicity associated with each of these products, an area that has been recently explored by our group (*Effah et al., 2025*).

Next, we performed hematoxylin and eosin (H&E) staining on the lung tissue sections to study the morphometric changes in the mouse lungs upon exposure to differently flavored e-cig aerosols. We did not find much evidence of tissue damage or airspace enlargement upon acute exposures in our model, as expected. However, we found evidence of increased alveolar septa thickening in the lungs of both male and female mice exposed to fruit-flavored e-cig aerosols (*Figure 1C*). This could be a result of lower agarose inflation observed for the mouse lungs in this group. We had challenges/difficulties with inflating these mouse lungs. Owing to the lack of proper inflation, the lungs were in a

**Table 1.** The levels of common elements found in the flavored e-liquids and e-cig aerosols as measured using inductively coupled plasma mass spectrometry (ICP-MS).

| Element | E-liquid; ppb/mg of e-liquid | | | | E-cig aerosol (mean ± SD); ppb/mg of e-liquid | | | |
|---|---|---|---|---|---|---|---|---|
| | PG:VG | Fruit | Menthol | Tobacco | PG:VG | Fruit | Menthol | Tobacco |
| S | 75.63 | 68.36 | 95.34 | 79.79 | 70.67 ± 16.94 | 74.05 ± 39.63 | 81.31 ± 27.24 | 91.81 ± 11.67 |
| Ni | 2.09 | 1.63 | 4.73 | 2.84 | 1.04 ± 1.00 | 30.47 ± 14.28 | 83.52 ± 20.53 | 41.73 ± 12.25 |
| Cu | 2.16 | 0.87 | 2.72 | 2.00 | 0.37 ± 0.22 | 1.04 ± 0.48 | 1.85 ± 0.70 | 1.32 ± 0.39 |
| Si | 1.05 | 1.13 | 1.55 | 1.26 | 1.10 ± 0.14 | 1.09 ± 0.35 | 1.32 ± 0.28 | 1.33 ± 0.12 |
| K | 0.80 | 0.84 | 1.21 | 0.67 | 0.94 ± 0.13 | 0.99 ± 0.45 | 1.32 ± 0.28 | 0.51 ± 0.10 |
| Na | 0.39 | 0.40 | 1.10 | 0.63 | 0.65 ± 0.11 | 0.91 ± 0.17 | 1.70 ± 0.37 | 0.79 ± 0.11 |
| W | 0.48 | 0.23 | 0.24 | 0.14 | 0.46 ± 0.20 | 0.30 ± 0.18 | 0.28 ± 0.14 | 0.22 ± 0.05 |
| Zn | 0.32 | 0.36 | 0.68 | 1.94 | 0.18 ± 0.01 | 1.45 ± 0.48 | 2.06 ± 0.48 | 1.71 ± 0.42 |
| Ir | 0.28 | 0.47 | 0.27 | 0.11 | 1.19 ± 0.76 | 0.35 ± 0.29 | 0.19 ± 0.12 | 0.14 ± 0.05 |
| B | 0.14 | 0.02 | 0.02 | 0.02 | 0.07 ± 0.02 | 0.02 ± 0.01 | 0.02 ± 0.01 | 0.02 ± 0.01 |
| Ta | 0.14 | 0.34 | 0.24 | 0.13 | 0.73 ± 0.30 | 0.24 ± 0.09 | 0.18 ± 0.06 | 0.13 ± 0.02 |
| Hf | 0.13 | 0.18 | 0.13 | 0.07 | 0.43 ± 0.19 | 0.14 ± 0.05 | 0.09 ± 0.03 | 0.07 ± 0.01 |
| Mo | 0.08 | 0.01 | 0.01 | 0.01 | 0.02 ± 0.01 | 0.07 ± 0.01 | 0.15 ± 0.04 | 0.06 ± 0.02 |
| Pd | 0.07 | 0.14 | 0.11 | 0.06 | 0.30 ± 0.14 | 0.11 ± 0.06 | 0.07 ± 0.02 | 0.06 ± 0.01 |
| Pt | 0.05 | 0.06 | 0.05 | 0.07 | 0.16 ± 0.07 | 0.09 ± 0.05 | 0.08 ± 0.04 | 0.08 ± 0.03 |
| Zr | 0.04 | 0.01 | 0.02 | 0.02 | 0.02 ± 0.01 | 0.02 ± 0.01 | 0.01 ± 0.00 | 0.01 ± 0.00 |
| Sn | 0.02 | 0.06 | 0.05 | 0.04 | 0.06 ± 0.02 | 0.07 ± 0.02 | 0.03 ± 0.01 | 0.03 ± 0.01 |
| Mg | 0.01 | 0.01 | 0.04 | 0.03 | 0.02 ± 0.00 | 0.12 ± 0.03 | 0.12 ± 0.03 | 0.10 ± 0.02 |
| Al | 0.01 | 0.01 | 0.01 | 0.01 | 0.01 ± 0.00 | 0.01 ± 0.00 | 0.01 ± 0.00 | 0.01 ± 0.00 |
| Co | 0.01 | 0.01 | 0.01 | 0.01 | 0.01 ± 0.00 | 0.02 ± 0.00 | 0.03 ± 0.01 | 0.02 ± 0.00 |
| Nb | 0.01 | 0.00 | 0.00 | 0.00 | 0.01 ± 0.00 | 0.00 ± 0.00 | 0.00 ± 0.00 | 0.00 ± 0.00 |
| Ba | 0.01 | 0.01 | 0.01 | 0.01 | 0.01 ± 0.01 | 0.01 ± 0.00 | 0.01 ± 0.00 | 0.01 ± 0.00 |
| Re | 0.01 | 0.01 | 0.01 | 0.02 | 0.06 ± 0.04 | 0.03 ± 0.03 | 0.02 ± 0.01 | 0.02 ± 0.01 |

collapsed state, which could be a probable explanation for our histological observation. Since this was not the prime focus of our study, we did not conduct further experiments to confirm our speculations.

## Detailed map of cellular composition during acute exposure to e-cig aerosols reveals distinct changes in immune cell phenotypes

As the principal focus of this study was to identify the flavor-dependent and independent effects upon exposure to commercially available e-cig aerosols at the single-cell level, we performed the scRNA seq on the mouse lungs from exposed and control mice. After quality control filtering (*Figure 2—figure supplement 1A, B*), normalization and scaling, we generated scRNA seq profiles of 71,725 cells in total. Except for the PG:VG group, all the rest of the treatments had approximately similar cell viabilities, cell capture, and other quality assessments. However, for normalization, equal features/genes were used across all the groups for subsequent analyses. A detailed account of the cell number (single-cell capture) and gene features identified before and after filtering upon QC check of sequenced data is provided in *Supplementary file 1a*.

Uniform Manifold Approximation and Projection (UMAP) was used for dimensionality reduction and visualization of cell clusters. Cell annotations were performed based on the established cell markers

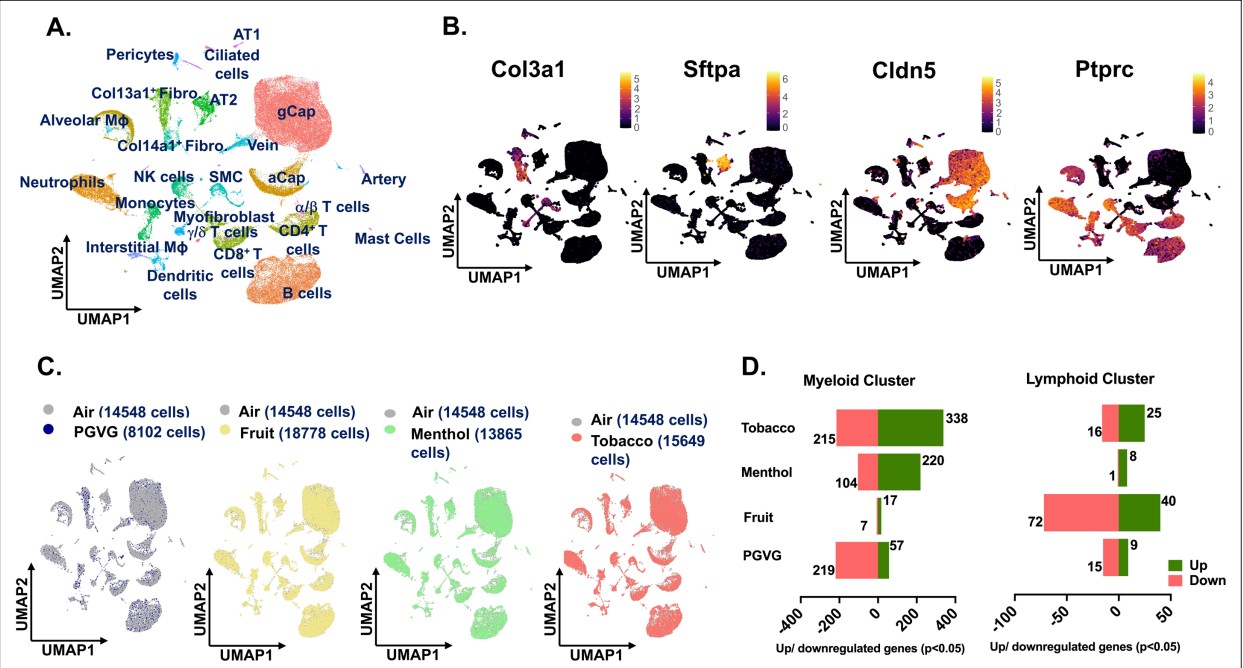

**Figure 2.** scRNA seq analyses reveal maximum changes in the transcriptional profile of immune cell population upon exposure to differently flavored e-cig aerosols. Male and female C57BL/6J mice (n = 2/sex/group) were exposed to 5-day nose-only exposure to differently flavored e-cig aerosols. The mice were sacrificed after the final exposure, and lungs from air (control) and differently flavored e-cig aerosol (fruit, menthol, and tobacco)-exposed mice were used to perform scRNA seq. Uniform Manifold Approximation and Projection (UMAP) plot of 71,725 cells captured during scRNA seq showing the 24 major cell clusters identified from control and experimental mouse lungs (**A**) and the expression of canonical markers used for identifying stromal (*Col3a1*), epithelial (*Sftpa1*), endothelial (*Cldn5*), and immune (*Ptprc*) cell populations. The intensity of expression is indicated by the black-yellow coloring (**B**). Group-wise comparison of the UMAPs upon comparing PG:VG (blue), fruit (yellow), menthol (green), and tobacco (red) versus air (gray) groups following dimensionality reduction and clustering of scRNA seq data (**C**). Bar plot showing the number of significant (p < 0.05) differentially up- (green) and downregulated (red) genes in myeloid and lymphoid clusters (*Figure 2—source data 1*) in PG:VG, fruit-, menthol-, and tobacco-flavored e-cig aerosol exposed mouse lungs as compared to air controls (**D**). Here, AT1: alveolar type I, AT2: alveolar type II, Fibro: fibroblast, M: macrophage, SMC: smooth muscle cell, gCap: general capillary, aCap: alveolar capillary, and NK: natural killer.

The online version of this article includes the following source data and figure supplement(s) for figure 2:

**Source data 1.** Number of significant (p < 0.05) differentially up- and downregulated genes in the myeloid and lymphoid clusters in PG:VG, fruit-, menthol-, and tobacco-flavored e-cig aerosol exposed mouse lungs when compared to air controls as plotted in *Figure 2D*.

**Figure supplement 1.** Quality check of the scRNA seq data generated using 10X Genomics pipeline.

**Figure supplement 1—source data 1.** Cell frequencies of major cell clusters (epithelial, endothelial, stromal, myeloid, and lymphoid) in control (air) and exposed (PG:VG, fruit, menthol, and tobacco) mouse lungs in each sample as determined by scRNA seq after filtering, clustering, and dimensionality reduction as plotted in *Figure 2—figure supplement 1C*.

in the Tabula Muris database and available published literature, and we identified 24 distinct cell clusters as shown in *Figure 2A*. The general clustering of individual cell types based upon the commonly known cell markers was used to identify **Endothelial** (identified by expression of *Cldn5*), **Epithelial** (identified by expression of *Sftpa1*), **Stromal** (identified by expression of *Col3a1*), and **Immune** (identified by expression of *Ptprc*) cell populations (*Figure 2B*). The 'FindVariableFeatures' from Seurat was used to identify cell-to-cell variation between the identified clusters, and the top 2000 variable genes identified in each cluster have been elaborated in *Supplementary file 1b*. We observed minor variations in the cell frequencies as observed through scRNA seq analyses within cell types across different treatment groups (*Figure 2—figure supplement 1C*). The largest proportion of cells was found in the endothelial cell cluster (43.87% of total per sample), followed by lymphoid (26.43% of total per sample) and myeloid (16.79% of total per sample) clusters. A detailed account of the two-way ANOVA statistics for the general clustering with cell types and treatment groups as independent variables is provided in *Supplementary file 1c*.

Groupwise comparisons of each treatment group versus air control did not show many changes in the cell clusters, thus showcasing little to no effect on the overall lung cellular compositions in treated

and control groups. However, the number of cells for the PG:VG group was very low (8102 cells) as compared to other treatments (~15,710 cells on average). We would like to report that this is an outcome of the low viability of this sample prior to scRNA seq, and not necessarily the effect of the treatment (*Figure 2C*). Differential gene expression analyses showed dysregulation of genes in all cell populations, but maximum effect was observed on the cells from immune cell clusters. This is not surprising, as the immune system, especially the myeloid cells, forms the frontline of the host's defenses against external stressors (*Del Fresno and Sancho, 2021*; *Marshall et al., 2018*). Compared to air, we observed a dysregulation of 553 genes (338 upregulated; 215 downregulated) in the myeloid cell cluster of mouse lungs exposed to tobacco-flavored e-cig aerosol. We identified 324 and 24 DEGs in the myeloid lung cluster from mouse lungs exposed to menthol- and fruit-flavored e-cig aerosols, respectively, as compared to air control (*Figure 2D*; *Supplementary file 1e–j*). For the lymphoid cluster, we observed maximum dysregulation in the lungs exposed to fruit-flavored e-cig aerosols with a total of 112 DEGs. In contrast, 41 and 9 significant DEGs were identified for the lymphoid cluster from lungs exposed to tobacco and menthol-flavored aerosols, respectively (*Figure 2D*; *Supplementary file 1l–q*).

It is important to mention here that most of the commercially available e-liquids/e-cig products use PG:VG as the base liquid to generate the aerosol and act as a carrier for flavoring chemicals. Thus, we compared the effect of PG:VG alone in our study to make further comparisons between individual flavoring products with that of PG:VG only. DESeq2 analyses showed dysregulation of 276 genes in mouse lungs exposed to PG:VG alone in the myeloid cell cluster as compared to air controls (*Figure 2D*; *Supplementary file 1d*). Contrary to this, exposure to PG:VG aerosols affected 24 genes in the lymphoid cluster as compared to air control (*Figure 2D*; *Supplementary file 1k*). Furthermore, when compared to PG:VG, exposure to fruit, menthol, and tobacco-flavored e-cig aerosol dysregulated 262, 873, and 960 genes in the myeloid cluster and 37, 64, and 112 genes in the lymphoid cluster, respectively. A detailed account of the DEGs identified upon comparing fruit, menthol, and tobacco-flavored e-cig aerosol exposed mouse lungs to PG:VG in myeloid and lymphoid clusters has been provided in *Supplementary file 1s–x*.

## Exposure to e-cig aerosols results in flavor-dependent changes in neutrophilic and eosinophilic response in vivo

To study the specific changes in the innate and adaptive immunity upon acute exposure to differently flavored e-cig aerosols, we first compared the changes in the overall cell population of individual cell types using scRNA seq. The scRNA seq data was validated with the help of flow cytometry using a larger cohort of animals. Since we observed sex-dependent variations in oxidative stress responses upon exposure to e-cig aerosols in previous studies (*Lamb et al., 2023*; *Wang et al., 2019*; *Lamb et al., 2022*), the flow cytometry data was analyzed in a sex-dependent fashion.

We did not observe any changes in the cell frequencies of alveolar macrophages across treatments through scRNA seq analyses. In general, there was a moderate decrease in the cell frequencies of alveolar macrophages in exposed mice as compared to air control independent of the flavor profile of the e-liquid employed (*Figure 3A*). Flow cytometric analyses confirmed little to no change in the alveolar macrophage (CD45$^+$ CD11b$^-$ SiglecF$^+$) percentages within the lung of exposed versus control mice (*Figure 3C, D*). Contrarily, we observed a flavor-dependent moderate increase in the cell frequencies of neutrophil clusters in menthol (0.1 ± 0.02) %- and tobacco (0.1 ± 0.06) %-flavored aerosol exposed mouse lungs as compared to controls (0.06 ± 0.02) % (*Figure 3B*). We used flow cytometry to validate the scRNA seq changes. Flow cytometry analyses showed an increase in the neutrophil (CD45$^+$ CD11b$^+$ Ly6G$^+$) percentages of menthol-flavored e-cig aerosol exposed mouse lung, corroborating with the scRNA seq results. Furthermore, our results showed this increase to be more pronounced in male mice (p = 0.0880) as compared to their female (p = 0.9662) counterparts (*Figure 3C, D*).

It is important to mention that due to the presence of nicotine in all the e-liquids used as treatments for this study, we added an extra control group: PG:VG +Nic for select experiments. Flow cytometric analyses revealed slight variations in the lung neutrophils and macrophage percentages observed in the PG:VG + Nic group as compared to the PG:VG only. However, none of these changes were significant. Furthermore, the patterns of change observed for the lungs exposed to aerosols from flavored e-liquids were quite distinct from those observed for PG:VG + Nic thus proving that the

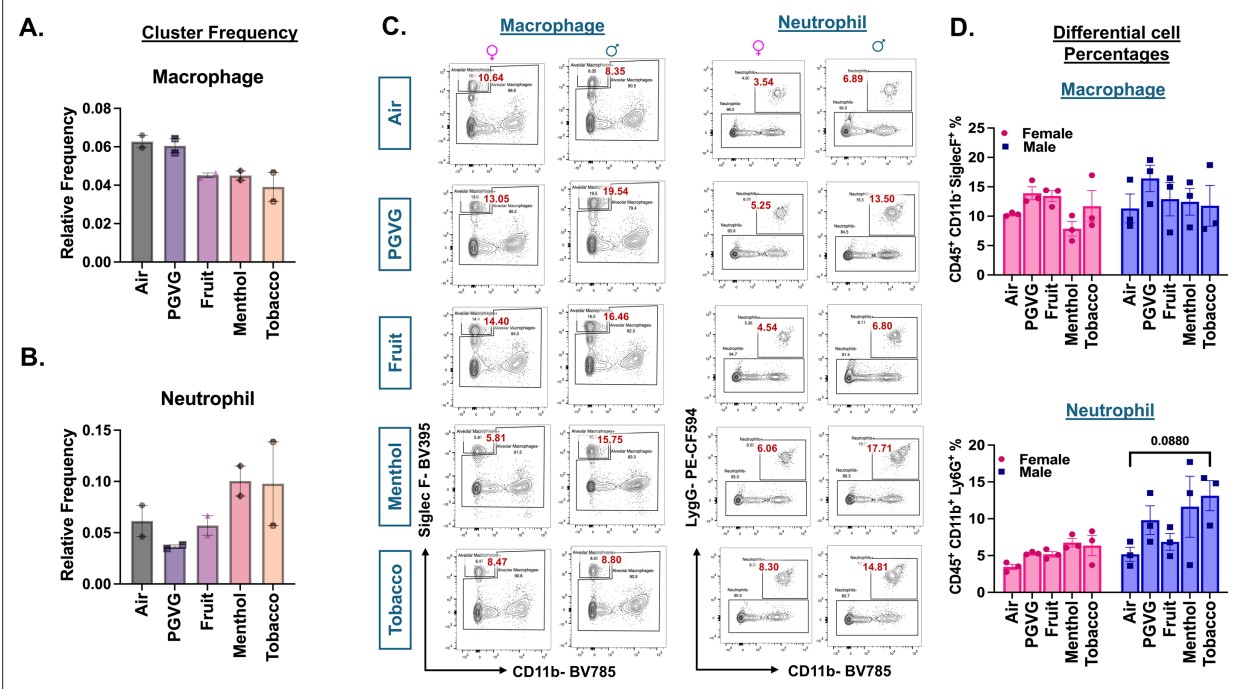

**Figure 3.** Cellular composition of myeloid cells in air and e-cig aerosol exposed mouse lungs reveals an increase in neutrophil count through scRNA seq and flow cytometry. Relative cell frequencies of alveolar macrophages (**A**) and neutrophils (**B**) across controls and flavored e-cig aerosol (***Figure 3—source data 1***) exposed mouse lungs as determined using scRNA seq. Representative flow plots (**C**) and bar graphs (**D**) showing sex-dependent changes in the percentages of neutrophils (CD45+ CD11b+ Ly6G+) and alveolar macrophage (CD45+ CD11b− SiglecF+) populations (***Figure 3—source data 2***) in lung digests from mice exposed to differently flavored e-cig aerosols. Values plotted and written in red on the flow plots are representative of the percentage of each cell population in the total CD45+ cells present in the lung homogenates from treatment and control groups. Data are shown as mean ± SEM (*n* = 3/sex/group). SE determined using two-way ANOVA with a Tukey post hoc test for all cell means, to analyze the main effects of sex and treatment and their interaction. The two-way ANOVA results are shown in ***Supplementary file 1c***.

The online version of this article includes the following source data and figure supplement(s) for figure 3:

**Source data 1.** Relative cell frequencies of alveolar macrophages and neutrophils across controls and flavored e-cig aerosol exposed mouse lungs as determined using scRNA seq as plotted in ***Figure 3A, B***.

**Source data 2.** Values showing the sex-dependent changes in the percentages of macrophages and neutrophils out of total CD45+ cells in lung digests from mice exposed to differently flavored e-cig aerosols as determined using flow cytometry as plotted in ***Figure 3D***, ***Figure 3—figure supplement 1A, B***.

**Figure supplement 1.** Inclusion of PG:VG + Nic group proves that nicotine is not the sole contributor to altered immune response in e-cig aerosol exposed mouse lungs.

**Figure supplement 2.** Gating strategy for the flow cytometry-based experiments.

observed changes are not solely due to the presence of nicotine in these groups (***Figure 3—figure supplement 1***).

Though we did not find a distinct eosinophil cluster in our scRNA seq analyzed data, interesting sex- and flavor-specific changes were observed in the lung eosinophil (CD45+ CD11b+/− CD11c− Ly6G− SiglecF+) population upon analyzing the flow cytometry results. A significant decline in the eosinophil percentage was observed in the lung homogenates from menthol (p = 0.0113) and tobacco (p = 0.0312)-flavored e-cig aerosol exposed male C57BL/6J mice as compared to air control. Contrarily, the eosinophil levels in the lung of male mice exposed to fruit-flavored e-cig aerosols did not show remarkable changes when compared to the levels found in control (***Figure 4***). These results further emphasize the need to study the sex-specific changes in the immune responses upon exposure to e-cig aerosols in vivo.

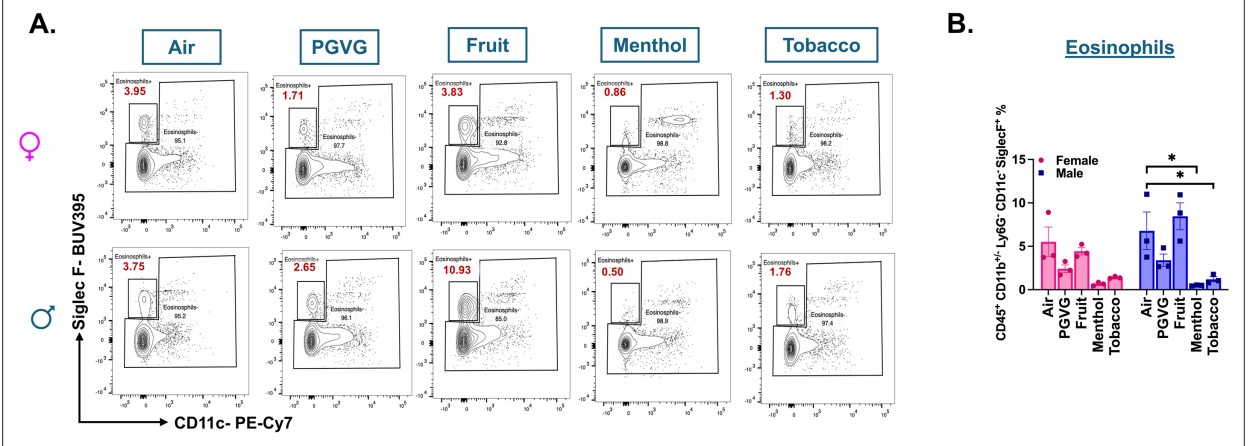

**Figure 4.** Flow cytometry analyses show significant decrease in the percentage of eosinophils in the lungs of menthol and tobacco-flavored e-cig aerosol exposed C57BL/6J mice. Representative flow plots (**A**) and bar graphs (**B**) showing the changes in the percentages of eosinophils (CD45$^+$ CD11b$^{+/-}$ CD11c$^-$ Ly6G$^-$ SiglecF$^+$) (**Figure 4—source data 1**) found in the lungs of differently flavored e-cig aerosol exposed mouse lungs as compared to air controls. Data are shown as mean ± SEM (*n* = 3/sex/group). *p < 0.05, per two-way ANOVA with a Tukey post hoc test for all cell means, to analyze the main effects of sex and treatment and their interaction. The two-way ANOVA results are shown in **Supplementary file 1c**. Values plotted and written in red on the flow plots are representative of the percentage of each cell population in the total CD45$^+$ cells present in the lung homogenates from treatment and control groups.

The online version of this article includes the following source data for figure 4:

**Source data 1.** Values showing the sex-dependent changes in the percentages of eosinophils out of total CD45$^+$ cells in lung digests from mice exposed to differently flavored e-cig aerosols as determined using flow cytometry as plotted in **Figure 4B**, **Figure 3—figure supplement 1C**.

## Activation of T-cell cytotoxic responses in lymphoid cells upon exposure to tobacco-flavored e-cig aerosols

We next studied the changes in the lymphoid clusters of treated and control samples. While we did not notice any change in the CD4$^+$ T-cell frequencies in treatment and control groups using scRNA seq, our flow cytometry data showed significant increase in the CD4$^+$ T cell (CD45$^+$ CD11c$^-$ Ly6G$^-$ CD11b$^-$ MHCII$^-$ CD4$^+$) frequencies in the lungs of tobacco-flavored e-cig aerosol exposed female (p = 0.0492) C57BL/6J mice (**Figure 5A, C, D**). scRNA data did not show any change in the cell frequencies of CD8$^+$ T cells in control and exposed mouse lungs either (**Figure 5B**). However, flow cytometric analyses presented a different picture. We found a significant sex-independent increase in the CD8$^+$ T (CD45$^+$ CD11c$^-$ Ly6G$^-$ CD11b$^-$MHCII$^-$CD8$^+$) cell percentages in the lungs of menthol- and tobacco-flavored e-cig aerosol exposed mice as compared to air control. Contrarily, the CD8$^+$ T-cell percentages increased in the fruit-flavored e-cig aerosol exposed male (p = 0.0163) mouse lungs and not in their female counterparts (**Figure 5C, D**). Of note, we did not observe any changes in the CD4$^+$ and CD8$^+$ T-cell percentages in the PG:VG + Nic group as compared to control, reiterating that the e-cig flavors are responsible for altered immune responses upon acute exposure in mice (**Figure 3— figure supplement 1**).

## Sub-clustering of myeloid cluster identifies a population of neutrophils devoid of Ly6G

To probe further, we subclustered the myeloid cell populations. Upon subclustering, we identified 14 unique clusters comprising all the major cell phenotypes including neutrophils, alveolar macrophages, interstitial macrophages, monocytes, dendritic cells, and mast cells (**Figure 6A**). A detailed account of all the cell types identified with their respective marker genes in myeloid subcluster is provided in **Supplementary file 1r**. On deeper evaluation, we identified two unique phenotypes of neutrophils (identified by *S100a8*, *S100a9*, *Il1b*, *Retnlg*, *Mmp9*, and *Lcn2*) in the mouse lungs. These clusters were named as 'Ly6G$^+$ Neutrophils' and 'Ly6G$^-$ Neutrophils' based on the presence or absence of Ly6G marker, respectively (**Figure 6B**). Ly6G is important for neutrophil migration, maturation, and function within the lung (**Lee et al., 2013**). Since we did not anticipate identifying such a population

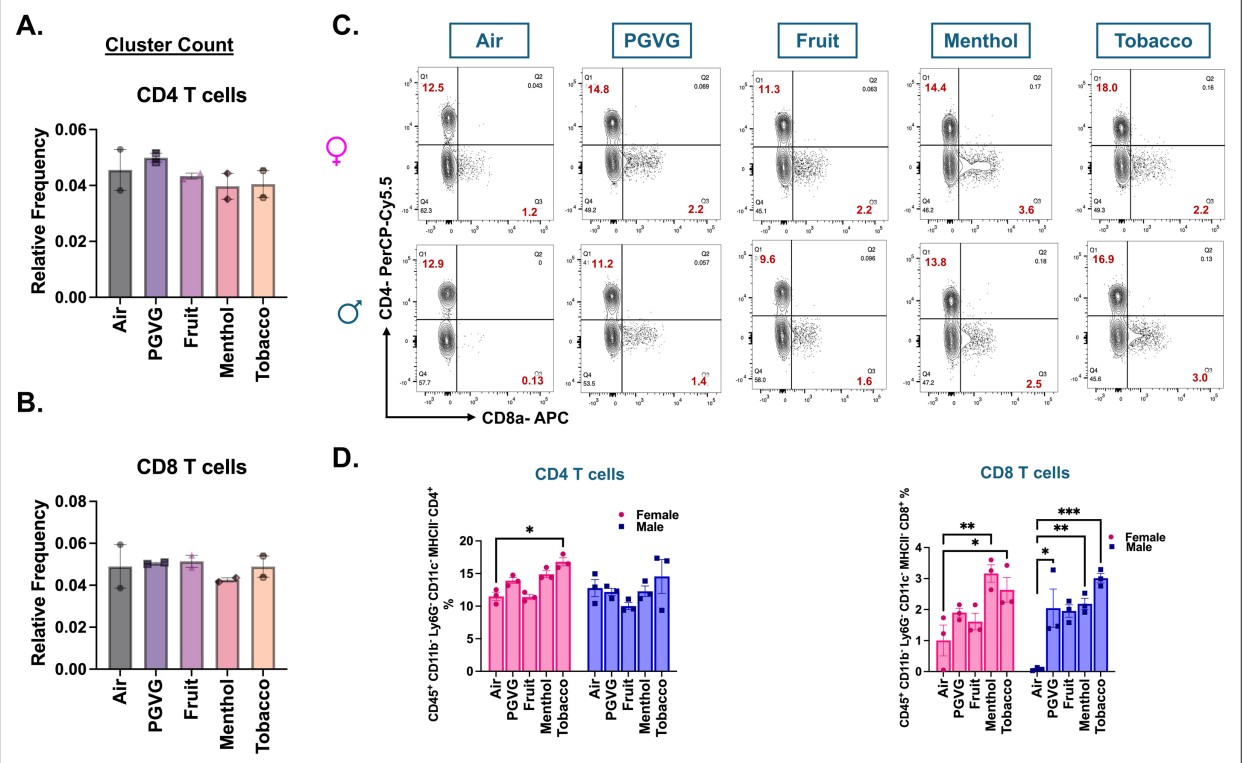

**Figure 5.** Flow cytometry results show sex-specific flavor-dependent increase in CD8+ T cells in lungs of differently flavored e-cig aerosol exposed C57BL/6J mouse. Relative cell frequencies of CD4+ (**A**) and CD8+ (**B**) T cells (**Figure 5—source data 1**) across controls and flavored e-cig aerosol exposed mouse lungs as determined using scRNA seq. Representative flow plots (**C**) and bar graph (**D**) showing changes in the mean cell percentages of CD4+ (CD45+ CD11c− Ly6G− CD11b− MHCII− CD4+) and CD8+ (CD45+ CD11c− Ly6G− CD11b− MHCII− CD8+) T cells (**Figure 5—source data 2**) in the lung tissue digests from male and female mice exposed to differently flavored e-cig aerosols as determined using flow cytometry. Values plotted and written in red on the flow plots are representative of the percentage of each cell population in the total CD45+ cells present in the lung homogenates from treatment and control groups. Data are shown as mean ± SEM (*n* = 3/sex/group). *$p < 0.05$, **$p < 0.01$, and ***$p < 0.001$; per two-way ANOVA with a Tukey post hoc test for all cell means, to analyze the main effects of sex and treatment and their interaction. The two-way ANOVA results are shown in **Supplementary file 1c**.

The online version of this article includes the following source data for figure 5:

**Source data 1.** Relative cell frequencies of CD4 and CD8 T cells across controls and flavored e-cig aerosol exposed mouse lungs as determined using scRNA seq as plotted in **Figure 5A, B**.

**Source data 2.** Values showing the sex-dependent changes in the percentages of CD4 and CD8 T cells out of total CD45+ cells in lung digests from mice exposed to differently flavored e-cig aerosols as determined using flow cytometry as plotted in **Figure 5D**, **Figure 3—figure supplement 1D, E**.

of neutrophils following acute e-cig exposure, we did not gather flow cytometry evidence to validate this finding.

We speculated that the Ly6G− population is a population of immature neutrophils. To confirm this possibility, we performed co-immunofluorescence using S100A8 (red; marker for neutrophil activation) and Ly6G (green) specific antibodies for control (air) and tobacco-flavored e-cig aerosol exposed mouse lungs. Co-immunofluorescence results showed a moderate increase (p = 0.4429) in the level of Ly6G+ cells in the lungs of tobacco-flavored aerosol exposed mice as compared to air. However, the level of S100A8+ cells decreased markedly (p = 0.0571) in e-cig exposed group, thus showing that activation of neutrophils could be affected upon acute exposure to e-cigs (**Figure 6—figure supplement 1C, D**).

## Exposure to fruit-flavored e-cig aerosols affects the mitotic pathway genes in the lymphoid cluster

To assess the effects of acute exposures to differently flavored e-cig aerosols on the cell composition in the mouse lungs, we performed differential expression analyses for each flavor in comparison to

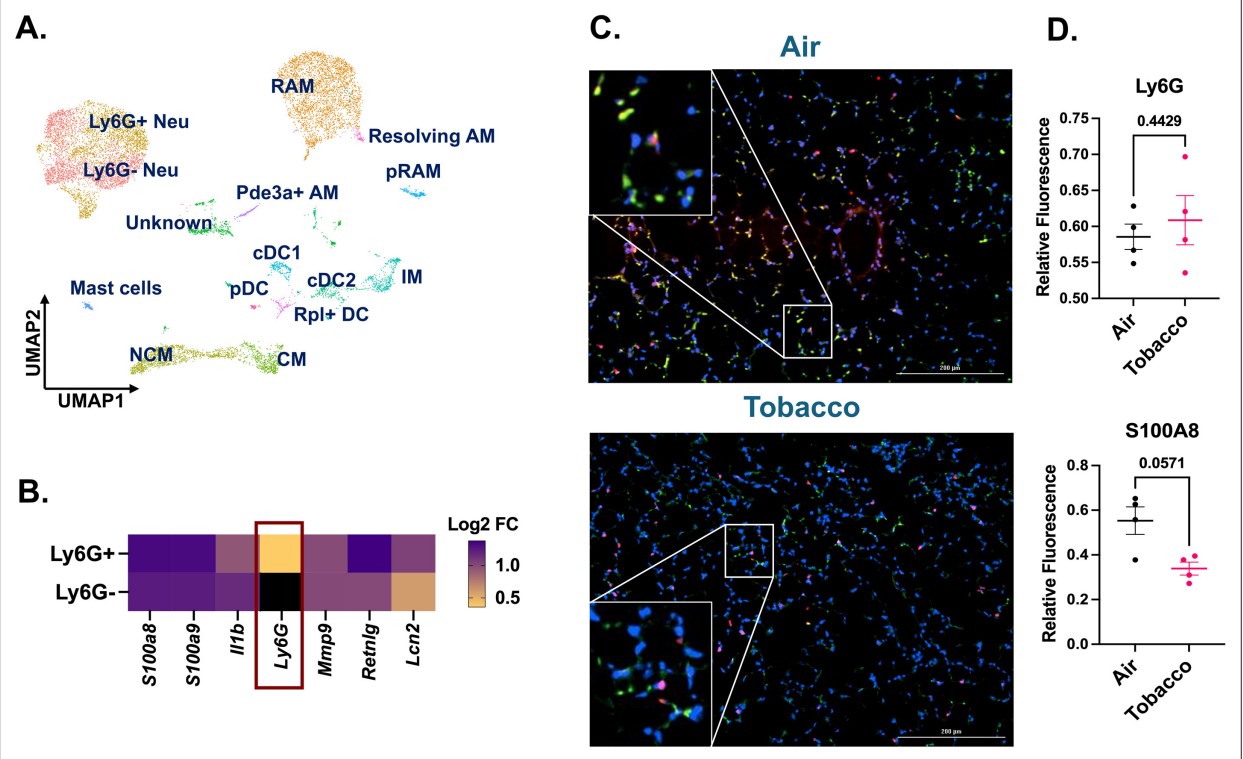

**Figure 6.** Co-immunofluorescence validates the increase of Ly6G⁻ and Ly6G⁺ neutrophil population in the tobacco-flavored e-cig exposed female C57BL/6J mice. The myeloid cell clusters were subsetted to identify two populations of neutrophils with and without the presence of Ly6G cell marker representing mature and immature neutrophils, respectively. Uniform Manifold Approximation and Projection (UMAP) showing the 14 distinct cell populations identified upon subsetting and re-clustering the myeloid clusters from scRNA seq dataset from control and e-cig exposed mouse lungs (**A**). Marker plot showing the differential expression of highly expressed genes in the Ly6G⁺ and Ly6G⁻ neutrophil cluster. The intensity of expression is indicated by the yellow-blue coloring; black represents nil value for expression for that gene (**B**). scRNA seq findings for presence of mature (Ly6G⁺) and immature (Ly6G⁻) neutrophils were validated by staining the tissue sections from tobacco-flavored e-cig aerosols and control (air) with Ly6G (green) and S100A8 (red, neutrophil activation marker). Representative images showing the co-immunostaining of Ly6G and S100A8 (shown as yellow puncta) at 20X magnification (**C**) with respective quantification of relative fluorescence for Ly6G and S100A8 (*Figure 6—source data 1*) (**D**) in control and tobacco-flavored e-cig aerosol exposed mice. Data are shown as mean ± SEM (*n* = 4/group). SE calculated per Mann–Whitney *U* test for pairwise comparisons. Here, Neu: neutrophil, AM: alveolar macrophage, RAM: resident AM, pRAM: proliferating RAM, IM: interstitial macrophage, CM: classical monocyte, NCM: non-classical monocyte, and DC: dendritic cell.

The online version of this article includes the following source data and figure supplement(s) for figure 6:

**Source data 1.** Mean value of relative fluorescence as determined from 6 to 10 random images captured from tissue sections from control and e-cig aerosol exposed mouse lungs quantitating Ly6G⁺ and S100A8⁺ puncta as plotted in *Figure 6D*.

**Figure supplement 1.** Co-immunofluorescence shows loss of S100A8 positive cells in tobacco-flavored e-cig aerosol exposed mouse lungs.

air and PG:VG controls. The fruit-flavored e-cig aerosol exposure had the mildest effect on the cell compositions and gene expression as compared to the controls in our study. We did not observe major dysregulation in the gene expression for myeloid cell cluster in fruit-flavored e-cig exposed mouse lungs as compared to air controls. A total of 24 genes (17 upregulated; 7 downregulated) were significantly ($p < 0.05$) differentially expressed in the myeloid clusters in the lungs of animals exposed to fruit-flavored e-cig aerosols (*Figure 7Ai*, *Supplementary file 1e*). GO analyses of differentially expressed upregulated genes showed enrichment of terms like 'NK cell-mediated immune response to tumor cells' (fold enrichment = 9.75; $p_{adj}$ = 0.0003), 'haptoglobin binding' (fold enrichment = 7.76; $p_{adj}$ = 0.0096) and 'transmembrane-ephrin receptor activity' (fold enrichment = 7.05; $p_{adj}$ = 0.014) in (*Figure 7Aii*, *Supplementary file 1f*) in fruit-flavored aerosol exposed mice lung as compared to control. Importantly, few gene clusters (*Hbb-bs*, *Hbb-bt*, *Hba-a2*, and *Hba-a1*) showed sex-specific changes in the gene expressions in exposed lungs as compared to control. These gene clusters are enriched for 'erythrocyte development' and warrant further study.

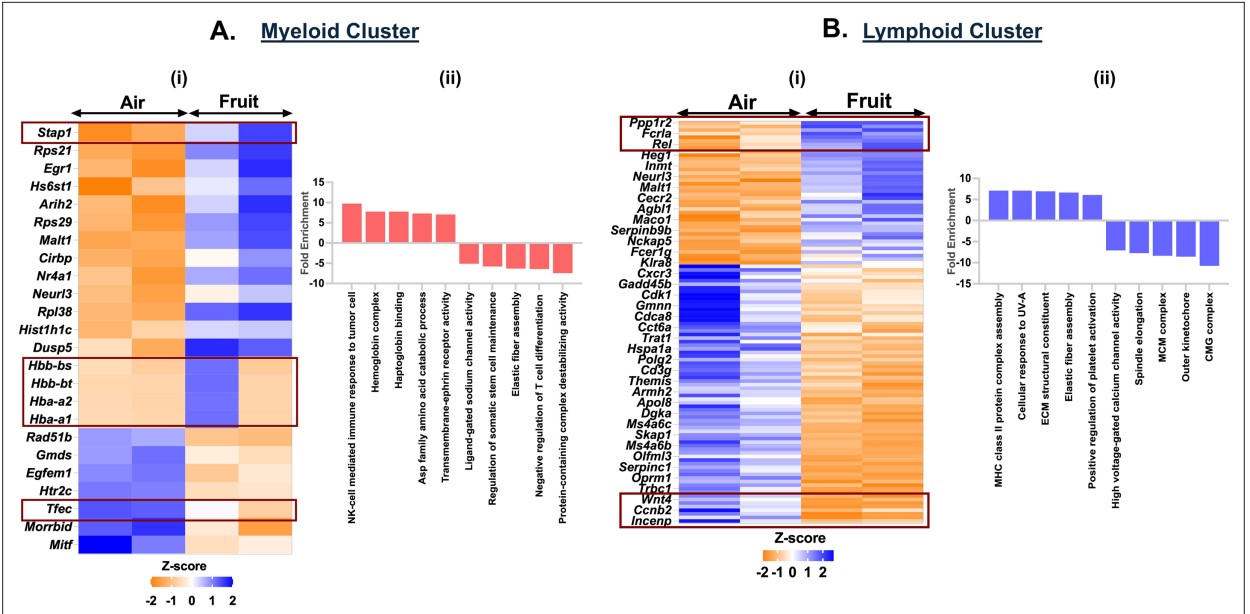

**Figure 7.** Exposure to fruit-flavored e-cig aerosols results in activation of oxidative stress-mediated innate immunity in C57BL/6J mouse lungs. Male and female C57BL/6J mice were exposed to 5-day nose-only exposure to fruit-flavored e-cig aerosols. The mice were sacrificed after the final exposure, and mouse lungs from air (control) and aerosol (fruit-flavored) exposed groups were used to perform scRNA seq. Heatmap and bar plot showing the DESeq2 (i) and GO analyses (ii) results from the significant ($p < 0.05$) up/downregulated differentially expressed genes (DEGs) in the myeloid (**A**) and lymphoid (**B**) cell cluster (*Figure 7—source data 1*) from fruit-flavored e-cig aerosol exposed mouse lungs as compared to controls. Data is representative of n = 2/sex/group.

The online version of this article includes the following source data for figure 7:

**Source data 1.** *Z*-score table showing significant ($p < 0.05$) differentially expressed genes (DEGs) and top 10 GO terms associated with the respective genes in the myeloid and lymphoid cluster from mouse lungs exposed to 5-day nose-only exposure to fruit-flavored e-cig aerosol when compared to air control as plotted in *Figure 7A(i, ii), B(i, ii)*.

We further found dysregulation of 112 genes (40 up- and 72 downregulated) in the lymphoid cluster of exposed mice. Upregulation of *H2-DMb2*, *H2-DMb1*, *Gp5*, *Pdpn*, among others, enriched for 'MHC class II protein complex assembly' (fold enrichment = 7.10; $p_{adj}$ = 0.0009) and 'positive regulation of platelet activation' (fold enrichment = 6.08; $p_{adj}$ = 0.0023) in exposed group as compared to air control. We also observed downregulation of genes like *Mcm4*, *Mcm2*, *Kif4*, *Cdca8*, *Cacna1f*, and *Cacnb3*, in the lymphoid cluster of mice exposed to fruit-flavored e-cig aerosol. GO analyses of the downregulated genes enriched for terms like 'CMG complex' (fold enrichment = 10.75; $p_{adj}$ = 3.39E-07), 'spindle elongation' (fold enrichment = 7.72; $p_{adj}$ = 0.0005) and 'high voltage-gated calcium channel activity' (fold enrichment = 7.09; $p_{adj}$ = 0.0059) in fruit-flavored aerosol exposed mouse lungs as compared to air controls (*Figure 7Bi–ii*, *Supplementary file 1l-m*).

## Exposure to menthol-flavored e-cig aerosols affects the immune cell function

We demonstrated an upregulation of 220 genes and a downregulation of 104 genes in the myeloid cluster of mouse lungs exposed to menthol-flavored e-cig aerosol as compared to ambient air. We observed increased expression of inflammatory genes including *Il12b*, *Arid5a*, *Il12a*, *Il1b*, *Cacna1d*, and *Cacnb2* enriching for terms like 'T-helper 1 cell cytokine production' (fold enrichment = 6.21; $p_{adj}$ = 0.0115) and 'L-type voltage-gated calcium channel complex' (fold enrichment = 5.58; $p_{adj}$ = 0.0191) in the myeloid cluster of menthol-flavored e-cig aerosol exposed mouse lungs as compared to air. We also found a downregulation in the expression of *Ovol1*, *Mapk15*, *Erbb2*, *Nrg4*, *Katnal2*, *Hspa1b*, and *Hspa1a* in the myeloid cells of flavored e-cig aerosols. GO analyses of the DEGs, thus, showed enrichment of terms like 'regulation of meiotic cell cycle phase transition' (fold enrichment = 6.48; $p_{adj}$ = 0.0051), 'ERBB4 signaling pathway' (fold enrichment = 5.40; $p_{adj}$ = 0.025), and 'protein-containing

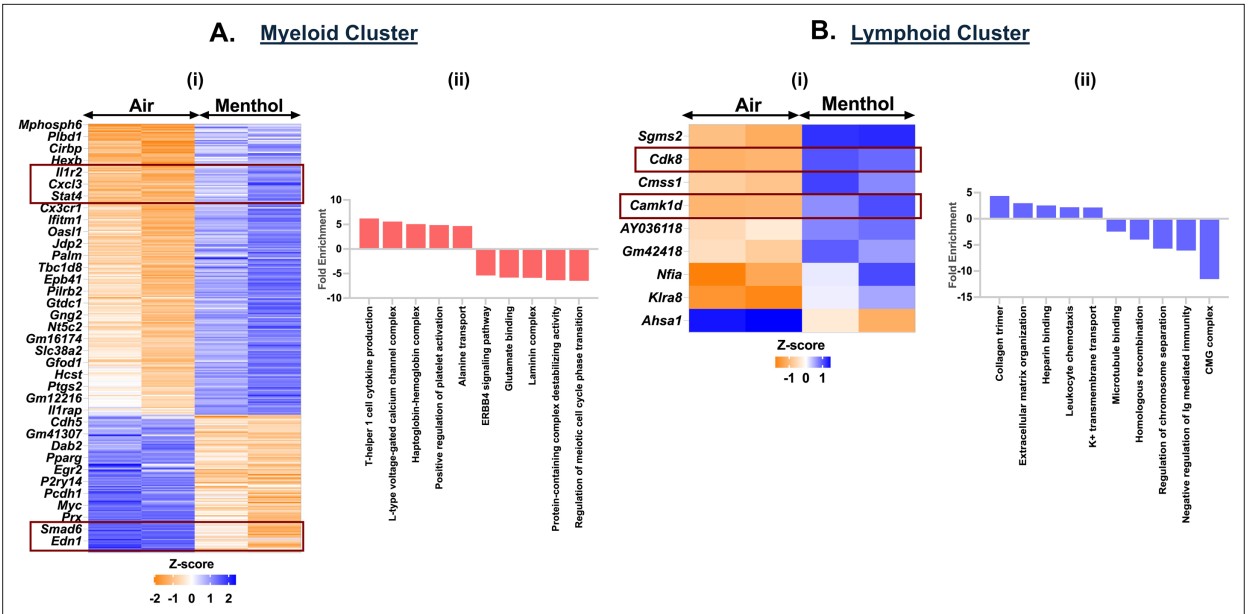

**Figure 8.** Exposure to menthol-flavored e-cig aerosols results in activation of innate immune responses in C57BL/6J mouse lungs. Male and female C57BL/6J mice were exposed to 5-day nose-only exposure to menthol-flavored e-cig aerosols. The mice were sacrificed after the final exposure, and mouse lungs from air (control) and aerosol (menthol-flavored) exposed groups were used to perform scRNA seq. Heatmap and bar plot showing the DESeq2 (i) and GO analyses (ii) results from the significant (p < 0.05) up/downregulated differentially expressed genes (DEGs) in the myeloid (**A**) and lymphoid (**B**) cell cluster (*Figure 8—source data 1*) from menthol-flavored e-cig aerosol exposed mouse lungs as compared to controls. Data is representative of n = 2/sex/group.

The online version of this article includes the following source data for figure 8:

**Source data 1.** *Z*-score table showing significant (p < 0.05) differentially expressed genes (DEGs) and top 10 GO terms associated with the respective genes in the myeloid and lymphoid cluster from mouse lungs exposed to 5-day nose-only exposure to menthol-flavored e-cig aerosol when compared to air control as plotted in *Figure 8A(i, ii), B(i, ii)*.

complex destabilizing activity ' (fold enrichment = 6.36; $p_{adj}$ = 0.0095) as the top hits as shown in *Figure 8Ai, ii* (*Supplementary file 1g, h*).

Contrary to the responses observed for exposure to fruit-flavored e-cig aerosols, we found significant (p < 0.05) upregulation in the expression of *Cdk8* and *Camk1d* genes in the lymphoid cell population for menthol-flavored aerosol exposed mouse lungs (*Figure 8Bi, Supplementary file 1n*). *Cdk8* (cyclin-dependent kinase 8) is a transcriptional regulator that has a role in the cell cycle progression (*Dannappel et al., 2018*). Whereas *Camk1d* (calcium/calmodulin-dependent protein kinase ID) functions to regulate calcium-mediated granulocyte function and respiratory burst within the cells (*Jin et al., 2022*). GO analyses of up- and downregulated genes showed enrichment of terms including 'regulation of toll-like receptor 9 signaling pathway' (fold enrichment = 5.96; $p_{adj}$ = 0.034), 'transforming growth factor beta activation' (fold enrichment = 5.42; $p_{adj}$ = 0.044), 'mitotic DNA replication' (fold enrichment = –7.98; $p_{adj}$ = 2.51E–05) and 'outer kinetochore' (fold enrichment = –8.58; $p_{adj}$ = 1.13E–08) in the lymphoid cluster from menthol-flavored e-cig aerosol exposed mouse lungs (*Figure 8Bii, Supplementary file 1o*).

## Exposure to tobacco-flavored e-cig aerosol elicits immune response in myeloid cell and cell cycle arrest in lymphoid cell population

Like menthol-, tobacco-flavored e-cig aerosol also elicited a significant increase in the expression of 338 genes and a decrease in 215 genes as compared to air controls in the myeloid cell cluster. We observed an increase in the expression of chemokines like *Stat4*, *Il1b*, *Il1bos*, *Il18r1*, *Unc13d*, *Lgals9*, and *Nkg7* in the myeloid cells resulting in enrichment of terms like 'T-helper 1 cell cytokine production' (fold enrichment = 6.66; $p_{adj}$ = 0.0011) and 'natural killer cell degranulation' (fold enrichment = 6.28; $p_{adj}$ = 0.0042) in tobacco-flavored e-cig aerosol exposed mouse lungs as compared to air controls.

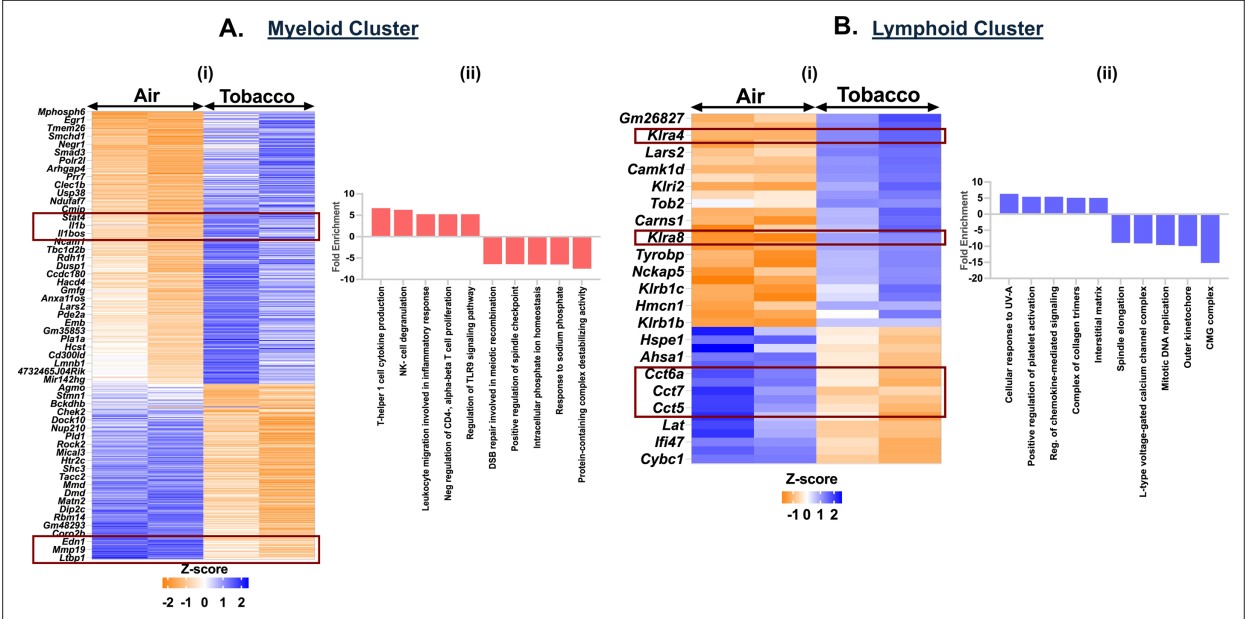

**Figure 9.** Exposure to tobacco-flavored e-cig aerosols results in activation of cytolysis and neutrophil chemotaxis in C57BL/6J mouse lungs. Male and female C57BL/6J mice were exposed to 5-day nose-only exposure to tobacco-flavored e-cig aerosols. The mice were sacrificed after the final exposure, and mouse lungs from air (control) and aerosol (tobacco-flavored) exposed groups were used to perform scRNA seq. Heatmap and bar plot showing the DESeq2 (i) and GO analyses (ii) results from the significant (p < 0.05) up/downregulated differentially expressed genes (DEGs) in the myeloid (**A**) and lymphoid (**B**) cell cluster from (*Figure 9—source data 1*) tobacco-flavored e-cig aerosol exposed mouse lungs as compared to controls. Data is representative of n = 2/sex/group.

The online version of this article includes the following source data and figure supplement(s) for figure 9:

**Source data 1.** Z-score table showing significant (p < 0.05) differentially expressed genes (DEGs) and top 10 GO terms associated with the respective genes in the myeloid and lymphoid cluster from mouse lungs exposed to 5-day nose-only exposure to tobacco-flavored e-cig aerosol when compared to air control as plotted in *Figure 9A(i, ii), B(i, ii)*.

**Figure supplement 1.** Exposure to flavored e-cig aerosol results in dysregulated chemokine signaling and T-cell activation.

**Figure supplement 1—source data 1.** Z-score values of the levels of cytokine/chemokine obtained in per mg of protein from mouse lung tissue samples following 5-day nose-only exposure to PG:VG, PG:VG + Nic, fruit, menthol, or tobacco-flavored e-cig aerosols as plotted in *Figure 9—figure supplement 1A*.

**Figure supplement 1—source data 2.** Z-score values of the gene expression of commonly dysregulated genes in the mouse lung tissue samples following 5-day nose-only exposure to PG:VG, fruit, menthol, or tobacco-flavored e-cig aerosols as compared to air controls as plotted in *Figure 9—figure supplement 1B*.

We further found downregulation of genes including *Mcmdc2*, *Rad51*, *Spp1*, and *Slc34a2* which enrich for terms like 'double-strand break repair involved in meiotic recombination' (fold enrichment = 6.44; $p_{adj}$ = 0.0019) and 'intracellular phosphate ion homeostasis' (fold enrichment = 6.52; $p_{adj}$ = 0.0052) in myeloid cluster from tobacco-exposed e-cig aerosols (*Figure 9Ai, ii*, *Supplementary file 1i, j*).

We also observed a downregulation of genes responsible for chaperone-mediated protein folding (*Cct5*, *Cct7*, and *Cct8*) in the lymphoid cluster from tobacco-flavored e-cig aerosol exposed mouse lungs. Downregulation of these genes could be indicative of the accumulation of misfolded proteins in these lungs which may lead to enhanced cell death (*Vallin and Grantham, 2019*; *Haeri and Knox, 2012*). DEG and GO analyses identified upregulation of *Robo1*, *Trem2*, *Padi2*, *Gp5*, *Gp9*, and *Pla2g4a* and downregulation of *Mcm4*, *Mcm2*, *Rad51*, *Cacnb2*, and *Cacnb3* in the lymphoid cluster from tobacco-flavored e-cig aerosol exposed mouse lungs resulting in enrichment of terms including 'regulation of chemokine-mediated signaling pathway' (fold enrichment = 5.43; $p_{adj}$ = 0.0304), 'positive regulation of platelet activation' (fold enrichment = 5.43; $p_{adj}$ = 0.0089), 'mitotic DNA replication' (fold enrichment = –9.6; $p_{adj}$ = 6.63E–07), and 'L-type voltage-gated calcium channel complex' (fold enrichment = –9.12; $p_{adj}$ = 0.0005) (*Figure 9Bi, ii*, *Supplementary file 1p, q*).

## Dysregulation of chemokine signaling and T-cell activation on exposure to flavored e-cig aerosols

Since we showed increased production of cytokines/chemokines, driving the immune responses in mouse lungs exposed to flavored e-cigs, we performed multianalyte assay to determine the levels of these inflammatory cytokines in the lung digests from the exposed animals as shown in *Figure 9— figure supplement 1A*. Exposure to tobacco-flavored e-cig aerosol resulted in a marked increase in the levels of chemotactic chemokines, including CXCL16, CXCL12, CXC3R, and proinflammatory cytokines, including CCl12, CCL17, CCL24, and Eotaxin, in the mouse lung digests as compared to air control. Interestingly, the fold changes in the PG:VG + Nic group were contrasting to those observed by PG:VG alone and flavored e-cig aerosol exposed mouse lungs, but none of these changes were highly significant.

To identify genes that were commonly altered upon exposure, we generated a list of common genes that were significantly dysregulated in exposure categories (fruit, menthol, and tobacco). We identified nine such target genes – *Neurl3*, *Egfem1*, *Stap1*, *Tfec*, *Mitf*, *Cirbp*, *Hist1h1c*, *Gmds*, and *Htr2c* – that were dysregulated in the myeloid cluster from lungs exposed to differently flavored e-cig aerosol, but not PG:VG. We observed significant upregulation of *Neurl3*, *Stap1*, *Cirbp*, and *Hist1h1c* and downregulation of *Tfec*, *Mitf*, *Gmds*, and *Htr2c* in the myeloid cluster of mice exposed to differently flavored e-cig aerosols. Upon analyzing the lymphoid cluster for commonly dysregulated genes, we identified – *Klra8* (Killer cell lectin-like receptor 8) and *Nfia* (nuclear factor I) – that were significantly upregulated in the exposure groups as compared to air-controls (*Figure 9—figure supplement 1B*). *Klra8* is a natural killer cell associated gene, and its upregulation is generally associated with viral infection associated host immune responses within the mouse lungs (*Lopes et al., 2022*; *Pommerenke et al., 2012*; *Akter et al., 2022*). *Nfia* is a transcriptional activator responsible for regulating Oxphos-mediated mitochondrial responses and proinflammatory pathways (*Kong et al., 2023*; *Hiraike et al., 2023*).

Overall, we identified a total of 29 commonly dysregulated gene targets that were identified from five major cell clusters and performed gene enrichment analyses on the identified targets to identify the top hits (*Tables 2 and 3*). Terms like 'negative regulation of immune system' (*Hmgb3/Gpam/Scgb1a1/Stap1/Ldlr*), 'positive regulation of lipid biosynthetic pathway' (*Htr2c/Gpam/Ldlr*), and 'receptor recycling' (*Ldlr/Ramp3*) were among the top hits in our observations (*Figure 9—figure supplement 1C*).

Of note, the data presented in this study is a sub-part of a larger study. In addition to the groups mentioned in this manuscript, we also had two additional groups of TDN and tobacco-free nicotine. Though further objectives and experimentations performed in both these studies were varied, common air and PG:VG samples were used for analyses of cytokine/chemokine and cell count data as described in our previous publication (*Lamb et al., 2023*).

## Discussion

E-cigs and associated products have constantly been under scrutiny by the US Food and Drug Administration (FDA) due to public health concerns. In February 2020, the FDA placed a regulation on all cartridge-based flavored e-cigs except for menthol and tobacco to reduce the use of e-cigs among adolescents and young adults. But it left a loophole for the sale of flavored (including menthol) disposable and open system e-cigs (*Ma et al., 2022*; *Sindelar, 2020*). Importantly, most e-cig-related bans in the US happened at the state level, thus allowing differential levels of restrictions imposed on the premarket tobacco applications and sales, which defeats the purpose of limiting their accessibility to the general public (*Bhalerao et al., 2019*; *Azagba et al., 2023*). In fact, the use of nicotine-containing e-cigs among youth and associated policy restrictions has recently been found to be linked to unintended increase in traditional cigarette use (*Cheng et al., 2025*). Each year, new products are introduced in the market with newer device designs and properties to lure the users (adults between the ages of 18–24 years), which makes it crucial to continue with the assessments of toxicity and health effects of e-cigs in an unbiased manner (*Kramarow and Elgaddal, 2023*).

Numerous studies indicate increased oxidative stress, DNA damage, and loss of neutrophil function due to exposure to e-cig aerosols in vitro and in vivo (*Muthumalage et al., 2019*; *Lamb et al., 2023*; *Lamb et al., 2020*; *Corriden et al., 2020*; *Jasper et al., 2024*; *Ren et al., 2022*). However, we

**Table 2.** List of top dysregulated genes on exposure to differently flavored (fruit, menthol, and tobacco) e-cig aerosol in C57BL/6J mouse lungs.

| Genes | Gene names | Gene function |
|---|---|---|
| Neurl3 | Neuralized E3 Ubiquitin Protein Ligase 3 | Ubiquitin protein ligase activity |
| Egfem1 | EGF-like and EMI domain containing 1 | Calcium ion-binding activity |
| Stap1 | Signal Transducing Adaptor Family Member 1 | Protein kinase binding and SH3/SH2 adaptor activity |
| Tfec | Transcription Factor EC | Multiple cellular processes including survival, growth, and differentiation |
| Mitf | Melanocyte Inducing Transcription Factor | Critical role in cell differentiation |
| Cirbp | Cold Inducible RNA Binding Protein | Role in cold-induced suppression of cell proliferation |
| Hist1h1c | H1.2 linker histone | Functions in the compaction of chromatin |
| Gmds | GDP-Mannose 4,6-Dehydratase | Coenzyme binding and NADP$^+$ binding |
| Htr2c | 5-Hydroxytryptamine Receptor 2C | G-protein-coupled receptor activity |
| Nfia | Nuclear Factor I A | DNA-binding transcription factor activity |
| Klra8 | killer cell lectin-like receptor | Carbohydrate binding activity. Acts upstream of or within response to virus |
| Trp53i11 | Tumor Protein P53 Inducible Protein 11 | Negative regulation of cell population proliferation |
| Ehd2 | EH Domain Containing 2 | Angiopoietin-like protein 8 regulatory pathway and response to elevated platelet cytosolic Ca$^{2+}$ |
| Ackr2 | Atypical Chemokine Receptor 2 | Recruitment of effector immune cells to the inflammation site |
| Marcks | Myristoylated Alanine Rich Protein Kinase C Substrate | Involved in cell motility, phagocytosis, membrane trafficking, and mitogenesis |
| Pfkl | Phosphofructokinase, | Protein binding and monosaccharide binding |
| Ramp3 | Receptor Activity Modifying Protein 3 | Signaling receptor activity and coreceptor activity |
| Chrm3 | Cholinergic Receptor Muscarinic 3 | Cellular responses such as adenylate cyclase inhibition, phosphoinositide degeneration, and potassium channel mediation |
| Sftpa1 | Surfactant Protein A1 | Carbohydrate binding and lipid transporter activity |
| Add3 | Adducin 3 | Actin binding and calmodulin binding |
| Hmgb3 | High Mobility Group Box 3 | Important role in maintaining stem cell populations and may be aberrantly expressed in tumor cells |
| Acot1 | Acyl-CoA Thioesterase 1 | Involved in acyl-CoA metabolic process; long-chain fatty acid metabolic process; and very long-chain fatty acid metabolic process |
| H1f0 | H1.0 Linker Histone | Cellular responses to stimuli and Programmed Cell Death |
| Scgb3a2 | Secretoglobin Family 3A Member 2 | Secreted lung surfactant protein |
| Scgb1a1 | Secretoglobin Family 1A Member 1 | Implicated in numerous functions including anti-inflammation, inhibition of phospholipase A2 and the sequestering of hydrophobic ligands |
| Gpam | Glycerol-3-Phosphate Acyltransferase, Mitochondrial | Acyltransferase activity and glycerol-3-phosphate *O*-acyltransferase activity |
| Cdh11 | Cadherin 11 | Integral membrane proteins that mediate calcium-dependent cell–cell adhesion |
| LDLR | Low-Density Lipoprotein Receptor | Cell surface proteins involved in receptor-mediated endocytosis of specific ligands |
| Myocd | Myocardin | Transcriptional co-activator of serum response factor (SRF) |

do not have much knowledge about the cell populations and biological signaling mechanisms that are most affected upon exposure to differently flavored e-cig aerosols at a single-cell level. To bridge this gap in knowledge, we studied the transcriptional changes in the inflammatory responses due to acute (1 hr nose-only exposure including 120 puffs for 5 consecutive days) exposure to fruit-, menthol-, and tobacco-flavored e-cig aerosols using single-cell technology. Short-term (2 hr/day for 3 or 5 consecutive days) e-cigarette exposure studies conducted by others and our group have shown an increase in

**Table 3.** Gene ontology results showing the top hits from the commonly dysregulated genes in all clusters on exposure to e-cig aerosols.

| GO ID | Ontology | Description | Gene ID | BgRatio | p.adjust |
|---|---|---|---|---|---|
| GO:0045907 | BP | Positive regulation of vasoconstriction | Htr2c/Chrm3/Add3 | 53/23,062 | 0.033815 |
| GO:0019229 | BP | Regulation of vasoconstriction | Htr2c/Chrm3/Add3 | 86/23,062 | 0.042491 |
| GO:1903978 | BP | Regulation of microglial cell activation | Stap1/Ldlr | 16/23,062 | 0.042491 |
| GO:0002683 | BP | Negative regulation of immune system process | Hmgb3/Gpam/Scgb1a1/Stap1/Ldlr | 464/23,062 | 0.042491 |
| GO:0010867 | BP | Positive regulation of triglyceride biosynthetic process | Gpam/Ldlr | 19/23,062 | 0.042491 |
| GO:0042310 | BP | Vasoconstriction | Htr2c/Chrm3/Add3 | 109/23,062 | 0.042491 |
| GO:0046889 | BP | Positive regulation of lipid biosynthetic process | Htr2c/Gpam/Ldlr | 110/23,062 | 0.042491 |
| GO:0010866 | BP | Regulation of triglyceride biosynthetic process | Gpam/Ldlr | 25/23,062 | 0.047046 |
| GO:0001919 | BP | Regulation of receptor recycling | Ldlr/Ramp3 | 29/23,062 | 0.047046 |
| GO:0007271 | BP | Synaptic transmission, cholinergic | Htr2c/Chrm3 | 29/23,062 | 0.047046 |
| GO:0150077 | BP | Regulation of neuroinflammatory response | Stap1/Ldlr | 29/23,062 | 0.047046 |
| GO:0090208 | BP | Positive regulation of triglyceride metabolic process | Gpam/Ldlr | 31/23,062 | 0.047046 |
| GO:0045987 | BP | Positive regulation of smooth muscle contraction | Chrm3/Myocd | 35/23,062 | 0.047046 |
| GO:0097242 | BP | Amyloid-beta clearance | Ldlr/Myocd | 36/23,062 | 0.047046 |
| GO:0040013 | BP | Negative regulation of locomotion | Htr2c/Mitf/Stap1/Myocd | 360/23,062 | 0.047046 |
| GO:0010667 | BP | Negative regulation of cardiac muscle cell apoptotic process | Acot1/Myocd | 37/23,062 | 0.047046 |
| GO:0001881 | BP | Receptor recycling | Ldlr/Ramp3 | 40/23,062 | 0.047046 |
| GO:0010664 | BP | Negative regulation of striated muscle cell apoptotic process | Acot1/Myocd | 40/23,062 | 0.047046 |
| GO:0019432 | BP | Triglyceride biosynthetic process | Gpam/Ldlr | 40/23,062 | 0.047046 |
| GO:0035296 | BP | Regulation of tube diameter | Htr2c/Chrm3/Add3 | 173/23,062 | 0.047046 |
| GO:0097746 | BP | Blood vessel diameter maintenance | Htr2c/Chrm3/Add3 | 173/23,062 | 0.047046 |
| GO:0001774 | BP | Microglial cell activation | Stap1/Ldlr | 42/23,062 | 0.047046 |
| GO:1903725 | BP | Regulation of phospholipid metabolic process | Htr2c/Ldlr | 42/23,062 | 0.047046 |
| GO:0035150 | BP | Regulation of tube size | Htr2c/Chrm3/Add3 | 174/23,062 | 0.047046 |

the pro-inflammatory cytokines and oxidative stress markers in both pulmonary and vascular tissues (*Wang et al., 2019*; *Kuntic et al., 2020*; *Wang et al., 2020c*). Since nose-only exposures are more direct than whole-body, we reduced the daily exposures of mice to 1 hr for 5 consecutive days. This dose and duration were found to show sex-dependent changes in MMP-2 and -9 activity and expression (*Lamb et al., 2023*). Prior literature formed the basis of the current study to assess the effect of acute exposure to differently flavored e-cig aerosols on the lung cell population using a nose-only exposure system.

Reports indicate that the release of metal ions due to the burning of metal coil is a major source of variation during e-cig exposures (*Zhao et al., 2020*; *Zhao et al., 2019*; *Alcantara et al., 2023*). A 2021 study reported the presence of 21 elements in the pod atomizers from different manufacturers identifying a high abundance of 11 elements including nickel, iron, zinc, and sodium among others

(*Omaiye et al., 2021*). Previous studies have also shown the presence of similar elements in the e-cig aerosols which could have a possible adverse health effect on the vapers (*Olmedo et al., 2018*; *Rastian et al., 2022*). More importantly, our study points toward a very important issue, which pertains to the product design of the e-cig vapes. We believe that the aerosol composition varies based on the type of atomizer, coil resistance, coil composition, and chemical reactivity of the e-liquid being used. While much work is done on the chemical composition of the flavors and e-liquids, the other aspects of device design remain understudied and must be an area of research in the future. In fact, a recent publication by our group emphasizes this aspect of product design by studying the exposure profile from low- and high-resistance coil in an e-cig (*Effah et al., 2025*). Another factor that may limit the interpretation of our results pertains to the correlation between the leached metal and the observed transcriptional changes. Since our study provides proof of day-to-day variation in the leaching of metal ions from the same liquid using the same atomizer in the emitted aerosols, it could be possible to develop a statistical model correlating the differential metal exposure to the gene expression changes to get a good assessment of the dose of each toxicant deposited on the lung surface, which has not been done in this study. This raises a pertinent question about what parameters should be controlled to design a comparative study between different flavors of e-cigs in the market per the standards of particle distribution and characteristics. Flavor-dependent variations in daily leaching of metals from an e-cig coil may result in variable deposition of toxicants onto the epithelial lining of both mouth and lungs in frequent vapers. Thus, future studies need to consider conditions like flavor, wattage, coil resistance, coil composition, and atomizer life when designing in vitro and in vivo studies to deduce the acute and chronic toxicities of vaping in humans.

E-cig vaping has been known to affect the innate and adaptive immune responses among vapers (*Jasper et al., 2024*; *Kalininskiy et al., 2021*; *Rebuli et al., 2021*; *Reidel et al., 2018*), but flavor-specific effects on immune function are not fully explored. While we expected to see flavor-dependent changes in our experiment, we did not anticipate observing interesting sex-dependent variation in the lung tissues using flow cytometry. In this respect, recent studies show concurring evidence suggesting sex-specific changes in lung inflammation, mitochondrial damage, gene expression, and even DNA methylation in mice exposed to e-cig aerosols (*Song et al., 2023*; *Chirumamilla et al., 2024*).

It is important to highlight here that while the changes in macrophage and neutrophil frequencies ascertained by scRNA seq corroborate with that observed using flow cytometry, similar correlation was not observed for the CD4$^+$ and CD8$^+$ T-cell data. The possible explanation for such a discrepancy could be high gene dropout rates in scRNA seq (*Qiu, 2020*), different analytical resolution for the two techniques (*Palit et al., 2019*), and pooling of samples in our single-cell workflow. Also, while we were able to identify some interesting changes in the eosinophil population within the lungs upon exposure to e-cigs using our flow cytometry data, we could not identify a cluster for eosinophils in our scRNA seq dataset. This could be due to the loss of this cell type during the sample preparation during scRNA seq capture or filtering out as empty droplets during data analyses. Theoretically, eosinophils are known to have lower RNA content and express fewer numbers of genes, which may result in them being identified as empty droplets and removed when running Cell Ranger (*Goss et al., 2025*). Despite the lack of data from scRNA seq, our findings are of relevance as an increased neutrophilic, but a decreased eosinophilic response is characteristic of severe inflammation, infection, and asthma (*Jukema et al., 2022*; *Flinkman et al., 2023*; *Lourda et al., 2021*). Further probing into the role of these two cell types in shaping the immune landscape within the lung upon e-cig aerosol exposure is paramount in understanding the specific toxicities and impact of each e-cig flavor on human health and well-being.

One of the most interesting discoveries from our single-cell analyses was the identification of a cluster of immature neutrophils without Ly6G surface marker. While we observed an increase in the cell percentages of these cells in our treatment groups, little to no change in the gene expression was noted (data not shown), which could be indicative of impaired function of mature neutrophil population, a fate being reported by various previous studies pertaining to e-cig exposures (*Madison et al., 2019*; *Kalininskiy et al., 2021*). In fact, our study identified (a) a moderate increase in the neutrophil count in menthol- and tobacco-flavored e-cig aerosol treated mouse lungs through scRNA seq analyses, (b) corroboration of the findings from scRNA seq using flow cytometry with more pronounced increase in Ly6G$^+$ neutrophils for male menthol-flavored e-cig aerosol exposed mice, and (c) identification of decreased co-localization for Ly6G and S100A8 in the lungs of tobacco-flavored e-cig aerosols

as compared to air control through co-immunostaining. Ly6G is an important marker of neutrophil recruitment and maturation in mammalian cells and has been reported in relation to various bacterial and parasitic infections in previous studies (*Deniset et al., 2017*; *Kleinholz et al., 2021*). Of note, this cluster did not express *SiglecF* and had high expression of other neutrophil markers including *S100A8/9*, *Lcn2*, and *Il1b*, thus negating the possibility of eosinophils being misrepresented as 'Ly6G⁻ neutrophils' in our analyses.

S100A8 acts as damage-associated molecular patterns and is responsible for neutrophil activation and neutrophil extracellular trap (NET) formation (*Sprenkeler et al., 2022*; *Guo et al., 2021*). While scRNA seq results in our study identified expression of S100A8/A9 genes in both neutrophil clusters – Ly6G⁺ and Ly6G⁻, the expression of *Ly6G* was totally absent for clusters identified as Ly6G⁻. Co-immunoprecipitation results also showed expression of both S100A8 and Ly6G markers within the lungs of treated and control lungs, but the co-localization of the two markers was more prominent in the control lungs, thus pointing toward a possible shift in the neutrophil function and activity upon exposure to e-cig aerosols. We are not the first to report the Ly6G deficiency among neutrophils. Previous work by *Deniset et al., 2017* and *Kleinholz et al., 2021* has highlighted the importance of Ly6G deficiency in relation to infection (*Deniset et al., 2017*; *Kleinholz et al., 2021*). *Deniset et al., 2017* study reported the presence of two populations of neutrophils in the splenic tissue of *Streptococcus pneumoniae* infected mice, based on the Ly6G expression – Ly6G$^{high}$ and Ly6G$^{intermediate}$. They found that while the former corresponds to mature neutrophils with ability of tissue migration and bacterial clearance, the latter constitutes the immature, immobile pool of neutrophils responsible for proliferation and replenishment of the mature pool of neutrophils (*Deniset et al., 2017*). The later study by *Kleinholz et al., 2021* demonstrated decreased uptake of *Leishmania major* in Ly6G-deficient mice, thus leading to delayed recruitment to and pathogen capture by neutrophils at the site of infection. Taken together, these studies provide evidence for a protective/compensatory role of the loss of Ly6G in the neutrophil population (*Kleinholz et al., 2021*). This, in addition to the recent findings by *Jasper et al., 2024* where neutrophils from healthy volunteers demonstrated a reduction in neutrophil chemotaxis, phagocytic function, and NET formation upon exposure to e-cig aerosols (*Jasper et al., 2024*), points toward a probable shift in the neutrophil dynamics upon exposure to e-cig aerosols. It is important to mention here that though our results from co-immunostaining using Ly6G and S100A8 pointed toward a shift in innate immune responses upon exposure to e-cig aerosols, it must not be confused with them being only expressed by neutrophil population. S100A8 is expressed in myeloid population including neutrophils, monocytes, and macrophages (*Averill et al., 2012*; *Wang et al., 2018*), and a subgroup of eosinophils is also known to express Ly6G (*Mair et al., 2021*). Thus, an in-depth characterization of these identified populations is important to understand the cellular and molecular responses toward e-cig aerosol exposure in vivo.

The myeloid and lymphoid limbs of immunity are interconnected (*Shanker and Marincola, 2011*; *Carroll and Prodeus, 1998*). In our study, we find an incidental shift in the neutrophil dynamics in menthol- and tobacco-flavored e-cig exposed mouse lungs. In contrast, we report an increase in the T-cell responses in the form of increased CD8⁺ T cells from both scRNA seq and flow cytometric analyses in male mice. In fact, increased expression of genes including *Malt1*, *Serpinb9b*, and *Sema4c* is indicative of enhanced T-cell-mediated immune response in the lungs of mice exposed to fruit (mango) flavored e-cig aerosols (*Wang et al., 2020d*; *Bird et al., 2014*; *Beland et al., 2014*). Contrary to this, exposure to menthol-flavored e-cig aerosols had a much milder effect on the lymphoid population within the lungs of C57BL/6J mice. We found evidence for increased lymphoid cell proliferation due to activation of cyclin-dependent protein kinase signaling mediated via expression of genes including *Cdk8* and *Camk1d* in these cells (*Jin et al., 2022*; *Szilagyi and Gustafsson, 2013*). Exposure to tobacco-flavored e-cig aerosol provided evidence for decreased chaperone-mediated protein folding, due to the downregulation of Chaperonin Containing TCP-1 (CCT) family of proteins. Chaperonin Containing TCP-1 proteins are important to regulate the production of native actin, tubulin, and other proteins crucial for cell cycle progression and cytoskeletal organization (*Brackley and Grantham, 2009*). This is in conjunction with the upregulation of *Klra4* and *Klra8* that is indicative of increased protein misfolding and cytotoxic responses in the lymphoid cells of tobacco-exposed e-cig aerosols (*Berry et al., 2013*; *Bolanos and Tripathy, 2011*).

Overall, we provide evidence of altered innate immune responses due to variable neutrophilic–eosinophilic function and increased T-cell proliferation and cytotoxicity in a flavor-dependent manner

upon exposure to e-cig aerosols in this study. An increase in the levels of CCL17, CCL20, CCL22, IL2, and Eotaxin in the lung digests from tobacco-exposed mouse lungs further supports this deduction as these cytokines/chemokines are associated with T-cell-mediated immune responses (*Israr et al., 2022*; *Li et al., 2020*; *Wang et al., 2024*; *Ross and Cantrell, 2018*; *Rapp et al., 2019*; *Jinquan et al., 1999*; *Chia et al., 2023*; *Böttcher et al., 2015*). Importantly, CXCL16 attracts T cells and natural killer cells to activate cell death. It is involved in LPS-mediated acute lung injury, an outcome which has also been linked with e-cig exposures in humans (*Tu et al., 2019*; *Christiani, 2020*).

Further, we compiled a list of commonly dysregulated genes in a flavor-independent manner and identified 29 gene targets. Signal-transducing adaptor protein-2 (*Stap1*), which is commonly upregulated upon exposure to e-cig aerosols, is known to regulate T-cell activation and airway inflammation, which is in agreement with the overall outcome of our findings (*Kagohashi et al., 2024*). Another gene that was found to be consistently dysregulated in many cell types was cold-inducible RNA binding protein (*Cirbp*). *Cirbp* is a stress response protein linked with stressors like hypoxia. Its upregulation upon e-cig exposure supports that vaping induces oxidative stress and can have adverse implications on the exposed cell types (*Zhu et al., 2024*). Importantly, this gene is involved in DNA repair mechanisms, thus making it crucial for cell survival pathways (*Firsanov et al., 2025*). 5-Hydroxytryptamine receptor 2C (*Htr2c*) and *Klra8* are other genes in this category of commonly dysregulated genes that are associated with enhancing inflammation and cell death (*Berry et al., 2013*; *Bolanos and Tripathy, 2011*; *Mikulski et al., 2010*). Overall, we provide a cell-specific resource of immune responses upon exposure to differently flavored e-cig aerosols.

Considering that scRNA technology has not been commonly used for e-cig research, ours is one of the first studies employing this technique to identify possible changes in the cellular composition and gene expressions. Importantly, we use the nose-only exposure system for our experiment to avoid exposure through other routes. However, despite the novel approach and state-of-the-art exposure system, we had a few limitations. First, we used a small sample size to identify the changes in the mouse lungs upon exposure to e-cig aerosols at a single-cell level. Due to the expensive nature of single-cell sequencing technology and limited information in the literature, we chose to design this experiment with small sizes of experimental and validation cohorts. But, based on the encouraging findings from this study, future studies could be designed with a larger sample size, longer durations of exposure, and more targeted approach to identify the acute and chronic effects of vaping in vivo. Second, we could not expand upon the sex-dependent changes observed through our work upon exposure. This was because such an effect was not anticipated when we conceived the idea of a short-term exposure in mice. However, considering the evidence from the current study, future experimental designs in our lab are considering sex as a crucial confounder for studying the effects of e-cig exposure in translational contexts. Third, the inclusion of PG:VG + Nic group was streamlined in this study, but in future work, the inclusion of this group for scRNA seq analyses to delineate the effects of nicotine alone on gene transcription is necessary. Fourth, we did not anticipate changes in the metal release on consecutive days of exposure at the start of our study. Later, our data pointed toward the importance of device design in e-cig exposures. Future studies need to identify the factors that may affect the daily composition of e-cig aerosols and devise a method of better monitoring these possible confounders. However, in this regard, our experiment does mimic the real-life scenario, as such variations due to prolonged storage of e-liquid and differences arising due to vape design must be common among human vapers.

In conclusion, this study identified cell-specific changes in the gene expressions characterized by altered neutrophil dynamics and accentuated T-cell cytotoxicity upon exposure to tobacco-flavored e-cig aerosols using single-cell technology. Furthermore, a set of top 29 dysregulated genes was identified that could be studied as markers of toxicity/immune dysfunction in e-cig research. Future work with larger sample sizes and sex distribution is warranted to understand the health impacts of long-term use of these novel products in humans.

## Methods
### Rigor and reproducibility statement
All experiments were designed to ensure rigor and reproducibility through inclusion of appropriate controls, randomization, blinding, and replication. Sample sizes and statistical analyses are described

in the Methods and figure legends. All reagents and analytical methods are reported in sufficient detail to allow replication. All the lab-based experiments comprise two technical replicates with a minimum of two to three biological replicates to ensure rigor. The data generated from this study is publicly available for future reference.

## Material availability statement

All materials used in this study are commercially available or publicly accessible, and no unique materials were generated. The respective catalog numbers and vendor information of the chemicals, kits, and/or antibodies used have been detailed in the respective sections.

## Animals

We ordered 5-week-old pups of male and female C57BL/6J mice (strain ID: 000664) from Jackson Laboratory to conduct this experiment. Prior to the start of the experiment, mice were housed at the URMC Vivarium for acclimatization. Thereafter, the animals were moved to the mouse Inhalation Facility at URMC for training and exposures.

One week prior to the start of the exposures, mice underwent a 5-day nose-only training to adapt themselves to the mesh restraints of the exposure tower. The mouse restraint durations were increased gradually to minimize the animal's stress and discomfort. Of note, the mouse sacrifice was performed within 8–12 weeks' age for each mouse group to ensure that the mouse age corresponds to the age of adolescents (12–17 years) in humans (*Dutta and Sengupta, 2016*; *Jackson et al., 2017*). Age and sex-matched animals ($n$ = 2/sex/group) used to perform scRNA seq were considered as the 'experimental cohort'; whereas another group of age and sex-matched ($n$ = 3/sex/group) mice exposed to air and flavored e-cig aerosol served as 'validation cohort' for this study.

## E-cigarette device and e-liquid

We utilized an eVic-VTC mini and CUBIS pro atomizer (SCIREQ, Montreal, Canada) with a BF SS316 1.0-ohm coil from Joyetech (Joyetech, Shenzhen, China) for vaping and the inExpose nose-only inhalation system from SCIREQ (SCIREQ, Montreal, Canada) for mouse exposures. Both air and PG:VG exposed mice groups were considered as controls for this experiment. We used commercially available propylene glycol (PG; EC Blend) and vegetable glycerin (VG; EC Blend) in equal volumes to prepare a 50:50 solution of PG:VG. For flavored product exposures, mice were exposed to three different e-liquids – a menthol flavor 'Menthol-Mint', a fruit flavor 'Mango' and a tobacco flavor 'Cuban Blend'. Of note, all the e-liquids were commercially manufactured with 50 mg/ml of TDN. So, all treatments have nicotine in addition to the flavoring mentioned, respectively. Additionally, we used a mixture of PG:VG with 50 mg/ml of TDN as a control for limited experiments to study the effect of nicotine alone in our treatment. This group is labeled PG:VG + Nic for the rest of the manuscript.

## E-cigarette exposure

Scireq Flexiware software with the InExpose Inhalation system was used for controlling the Joyetech eVic-VTC mini device to perform nose-only mouse exposures. For this exposure, we utilized a puffing profile that mimicked the puffing topography of e-cig users in two puffs per minute with a puff volume of 51 ml, puff duration of 3 s, and an inter-puff interval of 27 s with a 2-l/min bias flow between puffs (*Lee et al., 2018a*). *Figure 1—figure supplement 1A* depicts the experimental design and exposure system employed for this study.

Age-matched male and female ($n$ = 5 per sex) mice were used for each group, namely, air, PG:VG, fruit, menthol, and tobacco. To ensure rigor and reproducibility in our work, we have used age- and sex-matched control and treated mice in this study. Confounders like environment and stress were minimized by housing all the cages in environmentally controlled conditions and training all the mice (both control and treated) in nose-only chambers. Each group of mice was exposed to the above-mentioned puffing profile for 1 hr each day (120 puffs) for a total of five consecutive days. Additionally, a group of mice ($n$ = 3/sex) was exposed to PG:VG + Nic for the same duration using similar exposure profile to serve as control to assess the effect of nicotine on the observed changes using selected experiments. Air-exposed mice were exposed to the same puffing profile for a total of five consecutive days to ambient air. We recorded the temperature, humidity, and CO levels of the aerosols generated at the start, mid, and end of the exposure on each day using the Q-Trak Indoor Air

Quality Monitor 7575 (TSI, Shoreview, MN). Total particulate matter (TPM) sampling was done from the exhaust tubing of the setup at the 30 min mark of the exposure and at the inlet connected to the nose-only tower (shown in *Figure 1—figure supplement 1A*) immediately after the culmination of the exposure. Gravimetric measurements for TPM were also conducted to confirm relative dosage to each mouse group daily.

## Preparation of single-cell suspension

The animals (at 8–10 weeks' age) were sacrificed immediately after the final exposure. Vascular lung perfusion was performed using 3 ml of saline before harvesting the lung lobes for preparation of single-cell suspension. It is important to mention here that of the 5 lung samples/sex/group; 2/sex/group were used for histological assessments and scRNA analyses. Here, the left lung lobe was inflated using low-melting agarose and used for histology, while the rest of the uninflated lung lobes were used for preparing the single-cell suspension. We pooled the lung lobes for each sex per group for preparation of the single-cell suspension as depicted in *Figure 1A*. The lung lobes were weighed and digested using the Liberase method as described earlier (*Kaur, 2024*). Briefly, lung lobes were weighed and digested using Liberase (Cat# 5401127001; Roche, Basel, Switzerland) enzymatic cocktail with 1% DNase. The tubes were then transferred to the gentleMACS dissociator (Miltenyi, Gaithersburg, MD) and the manufacturer's protocol for mouse lung digestion was run. The sample tubes were next incubated at 37°C for 30 min with constant rotation, after which the suspension was strained through 70 μm MACS Smart Strainer. Thereafter, the suspension was centrifuged at $500 \times g$ for 10 min at 4°C, the supernatant was discarded, and 0.5 ml of RBC lysis buffer was added to the cell pellet to digest RBCs. The suspension was left on ice for 5 min in RBC lysis buffer and then 4 ml of ice-cold PBS with 10% FBS was added to stop the lysis. The suspension was again centrifuged at $500 \times g$ for 10 min at 4°C. The cell pellet was suspended in 1 ml PBS with 10% FBS, and cell number and viability were checked using AO/PI staining on a Nexcelom Cellometer Auto2000.

## Library preparation and single-cell sequencing

The prepared single-cell suspension was sent to the Genomics Research Center (GRC) at URMC for library preparation and single-cell sequencing. Library preparation was performed by control and treatment groups using the 10X single-cell sequencing pipeline by 10X Genomics, and 10,000 cells were captured per sample using the Chromium platform. The prepared library was sequenced on NovaSeq 6000 (Illumina, San Diego, CA) at a mean sequence depth of 30,000 reads per cell. Read alignment was performed to GRCm38 Sequence.

## Data analyses

We used the standard Seurat v4.3 analyses pipeline to analyze our data (*Hao et al., 2021*). In brief, the low-quality cells and potential doublets were excluded from the dataset to create the analyses dataset. The residual features due to the presence of RBCs were corrected before integration. 'scTransform' function was used for integration of all the datasets, after which the standard Seurat pipeline was used for data normalization of integrated data. 'FindVariableGenes' gene function was used to identify the variable genes for dimensionality reduction using PCA function. UMAP was used for dimensionality reduction and clustering of cells.

To identify the unique features and cell clusters within each cell subtype, we used the sub-setting feature within Seurat. After identifying the five major cell populations (epithelial, endothelial, stromal, myeloid, and lymphoid) in our datasets, each of these cell types was sub-clustered using the 'subset' function, normalized, and re-clustered. Cell annotation for each of the subsets was performed with the help of the Tabula Muris database (*The Tabula Muris Consortium et al., 2018*; *Hurskainen et al., 2021*). However, some clusters were annotated manually with the help of a literature search.

DESeq2 (V.1.42.1) was used to perform pseudobulk analyses to identify DEGs within each group. Here, genes showing a fold change >0.5 and <–0.5 along with a $p_{adj}$ value <0.05 were considered significantly dysregulated and plotted as heatmap using GraphPad. The ClusterProfiler R package (V. 4.10.1) (*Yu et al., 2012*) was employed to perform gene enrichment analyses of the DEGs (fold change >0.5 and <–0.5).

## Cytokine/chemokine assessment

We used a multiplex assay to determine the levels of cytokine/chemokine in the lung homogenates from control and e-cig aerosol exposed mouse lungs using commercially available Bio-Plex Pro Mouse Chemokine Assay (Cat# 12009159, Bio-Rad, Hercules, CA) per the manufacturer's instructions. Approximately 40 mg of mouse lung lobes were homogenized in 300 µl of 1X RIPA buffer with 0.1% protease and phosphatase inhibitor. The lung homogenate was stored on ice for 30 min. Following incubation, the homogenate was centrifuged at 15,000 rpm for 15 min at 4°C. The supernatant was collected and used for performing the multianalyte assay for determination of cytokine/chemokine levels using Luminex FlexMap3D system. A heatmap after normalization (Z-score) of the measured cytokine/chemokine to the protein amount loaded was plotted.

## Lung histology

The left lung lobe of mice used for scRNA seq was inflated with 1% low melting agarose and fixed with 4% neutral buffered PFA. Fixed lungs were dehydrated before being paraffin-embedded and sectioned (5 µm). H&E staining was performed by the Histology, Biochemistry, and Molecular Imaging Core at URMC. The H&E stain was observed at ×10 magnification using Nikon Elipse-Ni fluorescence microscope. Ten to fifteen random images were captured per sample.

## Flow cytometry

Flow cytometry was performed on the cells collected from lung homogenates from air and flavored e-cig aerosol exposed mouse lungs. For analyses of immune cell population in the lung, the lung lobes were digested as described earlier (*Kaur, 2024*). The single-cell suspension thus prepared was used to run flow cytometry using the BD LSRFortessa cell analyzer. Cells were blocked with CD16/32 (Tonbo biosciences 70-0161 u500, 1:10) to prevent nonspecific binding and stained with a master mix of Siglec F (BD OptiBuild Cat# 740280, 1:200), CD11b (Biolegend Cat# 101243, 1:200), Ly6G (BD Horizon Cat# 562700, 1:200), CD45 (Biolegend Cat# 103126, 1:200), CD11c (Biolegend Cat# 117318, 1:200), CD4 (Biolegend Cat# 116012, 1:200), and CD8 (eBiosciences Cat# 17-0081-82, 1:200). 7AAD (eBiosciences Cat# 00-6993-50, 1:10) was used as the nucleic acid dye to detect live and dead cells. The gating strategy used for this assay has been depicted in *Figure 3—figure supplement 2*.

## Metal analyses

To understand the levels of metals released during subsequent days of exposure, we performed ICP-MS on the e-cig aerosol condensates collected from each day of exposure using a Perkin Elmer ICP-MS model 2000C. The samples were run using a Total Quant KED protocol with 4 ml/min Helium flow and externally calibrated using a blank and a 100 ppb standard for the 51 elements. The samples were submitted to the Element Analyses facility at URMC, and levels of metals thus detected were plotted.

## Ly6G /S100A8 double staining

To determine the various populations of neutrophils in exposed and control groups, FFPE tissue sections from air and tobacco-flavored e-cig aerosol exposed lungs were stained with Ly6G and S100A8. In brief, 2–3 tissue sections per sample were deparaffinized using serial incubation in xylene followed by graded alcohol. Slides were incubated in 1X Citrate Buffer (Cat# S1699, Agilent, Santa Clara, CA) for 10 min at 95°C for antigen retrieval, which was followed by incubation at room temperature for 30 min. The slides were next washed with water and permeabilized using a permeabilization buffer (0.1% Triton-X in 1X TBST) for 10 min. Next, the slides were again washed with 1X TBST and blocked using Blocking buffer (5% goat serum in 1X TBST) for 30 min at room temperature. The blocked slides were incubated overnight at 4°C with Ly6G (Cat# 16-9668-85, Invitrogen, dilution: 1:100) and S100A8 (Cat# 26992-1-AP, Proteintech, dilution: 1:200). The next day, the slides were washed with 1X TBST and incubated for 2 hours at room temperature with goat anti-rabbit Alexa Fluor 594 (Cat# A11012, Invitrogen) and donkey anti-mouse Alexa Fluor 488 (Cat# A21202, Invitrogen) secondary antibody at 1:1000 dilution. Thereafter, the slides were washed and mounted with ProLong Diamond Antifade Mountant with DAPI (Cat# P36962, Invitrogen, Waltham, MA). Six to ten images were captured using Zoe Fluorescent Cell Imaging System (Bio-Rad Laboratories , Hercules, CA) at ×20 magnification. The

ImageJ deconvolution was used for quantifying the fluorescence in green ($Ly6G^+$) and red ($S100A8^+$) channels relative to DAPI (blue channel), and the relative fluorescence was plotted.

## Statistical significance

We used GraphPad Prism 10.5.0 for all statistical calculations. All the data plotted in this paper are expressed as mean ± SEM. Pairwise comparisons were done using unpaired *t* test while one-way analysis of variance (ANOVA) with ad hoc Tukey's test was employed for multi-group comparisons. To identify sex-based variations in our treatment groups, Tukey post hoc two-way ANOVA was employed.

## Code availability

Data collection was performed with mkfastq pipeline in Cell Ranger's (v7.0.1). Cell Ranger (v7.0.1) was used for cell and gene counting using the default settings. Single-cell analysis was performed using the Seurat R package (v4.3.0) using the recommended workflow.

# Acknowledgements

We would like to thank the Genomics Research Core, the Elemental Analyses Facility, and the Histology, Biochemistry, and Molecular Imaging Core at URMC for assisting us in the scRNA seq, metal analyses in aerosols, and lung sectioning and histology, respectively. We would also like to acknowledge Chengru Jiang for helping with histology image acquisition for this manuscript. This work was supported by WNY Center for Research on Flavored Tobacco Products (CRoFT) U54CA228110 and Toxicology Training Program T32 ES007026.

# Additional information

### Funding

| Funder | Grant reference number | Author |
| --- | --- | --- |
| National Institutes of Health | U54CA228110 | Irfan Rahman |
| National Institutes of Health | Toxicology Training Program T32ES007026 | Thomas Lamb |

The funders had no role in study design, data collection, and interpretation, or the decision to submit the work for publication.

### Author contributions

Gagandeep Kaur, Conceptualization, Data curation, Software, Formal analysis, Validation, Investigation, Visualization, Methodology, Writing – original draft, Writing – review and editing; Thomas Lamb, Data curation, Investigation, Methodology, Writing – original draft, Writing – review and editing; Ariel Tjitropranoto, Data curation, Software, Formal analysis, Methodology, Writing – original draft, Writing – review and editing; Irfan Rahman, Conceptualization, Resources, Data curation, Software, Formal analysis, Supervision, Funding acquisition, Validation, Investigation, Methodology, Writing – original draft, Project administration, Writing – review and editing

### Author ORCIDs

Gagandeep Kaur ⓘ https://orcid.org/0000-0002-2674-9947
Irfan Rahman ⓘ https://orcid.org/0000-0003-2274-2454

### Ethics

All experiments were conducted per the guidelines set by the University Committee on Animal Resources at the University of Rochester Medical Center (URMC). Care was taken to implement an unbiased and robust approach during the experimental design and conduct of each experiment to ensure data reproducibility per the National Institutes of Health standards.

Reviewer #1 (Public review): https://doi.org/10.7554/eLife.106380.4.sa1

Reviewer #3 (Public review): https://doi.org/10.7554/eLife.106380.4.sa2
Author response https://doi.org/10.7554/eLife.106380.4.sa3

## Additional files

### Supplementary files

Supplementary file 1. Detailed account of the analysed dataset and validation cohort for scRNA seq for e-cig exposed mouse lungs. (**a**) Table showing the QC parameters used before and after filtering and normalization of scRNA seq data. (**b**) Variable features with average $\log_2$ fold change and p value for genes in each cell cluster identified upon dimensionality reduction and clustering of 71,725 single cells from scRNA seq from treated and control samples. (**c**) Table showing the two-way ANOVA statistics for the cell frequencies identified through (a) scRNA seq analyses for general clustering using treatment and cell types as independent variables and (b) flow cytometric analyses using treatment and sex as independent variables. (**d**) (a) DESeq2 results showing the significant (p < 0.05) DEGs and (b) GO results of the DEGs in the myeloid cell cluster from mouse lungs exposed to 5-day nose-only exposure to PG:VG when compared to air control. (**e**) DESeq2 results showing the significant (p < 0.05) DEGs in the myeloid cell cluster from mouse lungs exposed to 5-day nose-only exposure to fruit-flavored e-cig aerosol when compared to air control. (**f**) GO analyses results of the (a) upregulated DEGs ($\log_2$ fold change >0.5) and (b) downregulated DEGs ($\log_2$ fold change <–0.5) in the myeloid cell cluster from mouse lungs exposed to 5-day nose-only exposure to fruit-flavored e-cig aerosols when compared to air control. (**g**) DESeq2 results showing the significant (p < 0.05) DEGs in the myeloid cell cluster from mouse lungs exposed to 5-day nose-only exposure to menthol-flavored e-cig aerosol when compared to air control. (**h**) GO analyses results of the (a) upregulated DEGs ($\log_2$ fold change >0.5) and (b) downregulated DEGs ($\log_2$ fold change <–0.5) in the myeloid cell cluster from mouse lungs exposed to 5-day nose-only exposure to menthol-flavored e-cig aerosols when compared to air control. (**i**) DESeq2 results showing the significant (p < 0.05) DEGs in the myeloid cell cluster from mouse lungs exposed to 5-day nose-only exposure to tobacco-flavored e-cig aerosols when compared to air control. (**j**) GO analyses results of the (a) upregulated DEGs ($\log_2$ fold change >0.5) and (b) downregulated DEGs ($\log_2$ fold change <–0.5) in the myeloid cell cluster from mouse lungs exposed to 5-day nose-only exposure to tobacco-flavored e-cig aerosols when compared to air control. (**k**) (a) DESeq2 results showing the significant (p < 0.05) DEGs and (b) GO results of the DEGs in the lymphoid cell cluster from mouse lungs exposed to 5-day nose-only exposure to PG:VG when compared to air control. (**l**) DESeq2 results showing the significant (p < 0.05) DEGs in the lymphoid cell cluster from mouse lungs exposed to 5-day nose-only exposure to fruit-flavored e-cig aerosol when compared to air control. (**m**) GO analyses results of the (a) upregulated DEGs ($\log_2$ fold change >0.5) and (b) downregulated DEGs ($\log_2$ fold change <–0.5) in the lymphoid cell cluster from mouse lungs exposed to 5-day nose-only exposure to fruit-flavored e-cig aerosols when compared to air control. (**n**) DESeq2 results showing the significant (p < 0.05) DEGs in the lymphoid cell cluster from mouse lungs exposed to 5-day nose-only exposure to menthol-flavored e-cig aerosol when compared to air control. (**o**) GO analyses results of the (a) upregulated DEGs ($\log_2$ fold change >0.5) and (b) downregulated DEGs ($\log_2$ fold change <–0.5) in the lymphoid cell cluster from mouse lungs exposed to 5-day nose-only exposure to menthol-flavored e-cig aerosols when compared to air control. (**p**) DESeq2 results showing the significant (p < 0.05) DEGs in the lymphoid cell cluster from mouse lungs exposed to 5-day nose-only exposure to tobacco-flavored e-cig aerosol when compared to air control. (**q**) GO analyses results of the (a) upregulated DEGs ($\log_2$ fold change >0.5) and (b) downregulated DEGs ($\log_2$ fold change <–0.5) in the lymphoid cell cluster from mouse lungs exposed to 5-day nose-only exposure to tobacco-flavored e-cig aerosols when compared to air control. Variable features with average $\log_2$ fold change and p value for genes in each cell cluster identified upon dimensionality reduction and clustering of myeloid subset. (**s**) DESeq2 results showing the DEGs in the myeloid cell cluster from mouse lungs exposed to 5-day nose-only exposure to fruit flavored e-cig aerosols when compared to PG:VG exposure. (**t**) DESeq2 results showing the DEGs in the myeloid cell cluster from mouse lungs exposed to 5-day nose-only exposure to menthol flavored e-cig aerosols when compared to PG:VG exposure. (**u**) DESeq2 results showing the DEGs in the myeloid cell cluster from mouse lungs exposed to 5-day nose-only exposure to tobacco-flavored e-cig aerosols when compared to PG:VG exposure. (**v**) DESeq2 results showing the DEGs in the lymphoid cell cluster from mouse lungs exposed to 5-day nose-only exposure to fruit-flavored e-cig aerosol when compared to PG:VG exposure. (**w**) DESeq2 results showing the DEGs in the lymphoid cell cluster from mouse lungs exposed to 5-day nose-only exposure to menthol-flavored e-cig aerosol when compared to PG:VG

exposure. (**x**) DESeq2 results showing the DEGs in the lymphoid cell cluster from mouse lungs exposed to 5-day nose-only exposure to tobacco-flavored e-cig aerosol when compared to PG:VG exposure.

MDAR checklist

## Data availability

The datasets generated are deposited on NCBI Gene Expression Omnibus under accession code GSE263903' The data will be publicly available upon publication or on February 1, 2026, whichever is earlier. All other relevant data supporting the key findings of this study are available within the article and its supplementary files. Source data are provided with this paper.

The following dataset was generated:

| Author(s) | Year | Dataset title | Dataset URL | Database and Identifier |
|---|---|---|---|---|
| Kaur G, Lamb T, Tjitropranoto A, Rahman I | 2026 | Single-cell transcriptomics identifies altered neutrophil dynamics and accentuated T-cell cytotoxicity in tobacco -flavored e-cigarette -exposed mouse lungs | https://www.ncbi. nlm.nih.gov/geo/ query/acc.cgi?acc= GSE263903 | NCBI Gene Expression Omnibus, GSE263903 |

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
