## [Editor Report · eLife Assessment]

This manuscript by Kaur et al. identifies differential gene expression in distinct cell populations, specifically myeloid and lymphoid cells, following short-term exposure to e-cigarette aerosols with various flavors. Their findings are **useful** because they provide a single-cell sequencing data resource for assessing which genes and cellular pathways could be affected by e-cig aerosols and their components. However, the evidence is **incomplete** due to limited number of biological replicates per condition, as well as due to the lack of in vivo validation.

---

## [Referee Report · Reviewer #1 (Public review)]

Summary:

The authors assess the impact of E-cigarette smoke exposure on mouse lungs using single-cell RNA sequencing. Air was used as control and several flavors (fruit, menthol, tobacco) were tested. Differentially expressed genes (DEGs) were identified for each group and compared against the air control. Changes in gene expression in either myeloid or lymphoid cells were identified for each flavor and the results varied by sex. The scRNAseq dataset will be of interest to the lung immunity and e-cig research communities, and some of the observed effects could be important. Unfortunately, the revision did not address the reviewers' main concerns about low replicate numbers and lack of validations. The study remains preliminary and no solid conclusions could be drawn about the effects of E-cig exposure as a whole or any flavor-specific phenotypes.

Strengths:

The study is the first to use scRNAseq to systematically analyze the impact of e-cigarettes on the lung. The dataset will be of broad interest.

Weaknesses:

This study had only N=1 biological replicates for the single-cell sequencing data per sex per group and some sex-dependent effects were observed. This could have been remedied by validating key observations from the study using traditional methods such as flow cytometry and qPCR, but the limited number of validation experiments did not support the conclusions of the scRNAseq analysis. An important control group (PG:VG) had extremely low cell numbers and therefore could not be used to derive meaningful conclusions. Statistical analysis is lacking in almost all figures. Overall, this is a preliminary study with some potentially interesting observations.

(1) The only new validation experiment for this revision is the immunofluorescent staining of neutrophils in Figure 4. The images are very low resolution and low quality and it is not clear which cells are neutrophils. S100A8 (calprotectin) is highly abundant in neutrophils but not strictly neutrophil-specific. It's hard to distinguish positive cells from autofluorescence in both ly6g and S100a8 channels. No statistical analysis is presented for the quantified data from this experiment.

(2) The relevance of Fig. 3A and B are unclear since these numbers only reflect the number of cells captured in the scRNAseq experiment and the biological meaning of this data is not explained. Flow cytometry quantification is presented as cell counts but percentage of cells from the CD45+ gate should be shown. No statistical analysis is shown, and flow cytometry results do not support the conclusions of scRNAseq data.

---

## [Referee Report · Reviewer #3 (Public review)]

This work aims to establish cell-type-specific changes in gene expression upon exposure to different flavors of commercial e-cigarette aerosols compared to control or vehicle. Kaur et al. conclude that immune cells are most affected, with the greatest dysregulation found in myeloid cells exposed to tobacco-flavored e-cigs and lymphoid cells exposed to fruit-flavored e-cigs. The up- and down-regulated genes are heavily associated with innate immune response. The authors suggest that a Ly6G-deficient subset of neutrophils is found to be increased in abundance for the treatment groups, while gene expression remains consistent, which could indicate impaired function. Increased expression of CD4+ and CD8+ T cells along with their associated markers for proliferation and cytotoxicity is thought to be a result of activation following this decline in neutrophil-mediated immune response.

Strengths:

Single-cell sequencing data can be very valuable in identifying potential health risks and clinical pathologies of lung conditions associated with e-cigarettes considering they are still relatively new.

Not many studies have been performed on cell-type-specific differential gene expression following exposure to e-cig aerosols.

The assays performed address several factors of e-cig exposure such as metal concentration in the liquid and condensate, coil composition, cotinine/nicotine levels in serum and the product itself, cell types affected, which genes are up- or down-regulated and what pathways they control.

Considerations were made to ensure clinical relevance such as selecting mice whose ages corresponded with human adolescents so that data collected was relevant.

The discussion addresses the limitations of this study.

Weaknesses:

The exposure period of 1 hour a day for 5 days is not representative of chronic use and this time point may be too short to see a full response in all cell types. There is no gold standard in the field.

Most findings are based on scRNA-seq alone, so interpretations should be made with care as some conclusions are observational.

This paper provides a good foundation for future follow-up studies that will examine the effects of e-cig exposure on innate immunity.

---

## [Author Response]

The following is the authors’ response to the previous reviews.

**Public Reviews:**

**Reviewer #1 (Public review):**
The authors assess the impact of E-cigarette smoke exposure on mouse lungs using single cell RNA sequencing. Air was used as control and several flavors (fruit, menthol, tobacco) were tested. Differentially expressed genes (DEGs) were identified for each group and compared against the air control. Changes in gene expression in either myeloid or lymphoid cells were identified for each flavor and the results varied by sex. The scRNAseq dataset will be of interest to the lung immunity and e-cig research communities and some of the observed effects could be important. Unfortunately, the revision did not address the reviewers' main concerns about low replicate numbers and lack of validations. The study remains preliminary, and no solid conclusions could be drawn about the effects of E-cig exposure as a whole or any flavor-specific phenotypes.Strengths:The study is the first to use scRNAseq to systematically analyze the impact of e-cigarettes on the lung. The dataset will be of broad interest.Weaknesses:scRNAseq studies may have low replicate numbers due to the high cost of studies but at least 2 or 3 biological replicates for each experimental group is required to ensure rigor of the interpretation. This study had only N=1 per sex per group and some sex-dependent effects were observed. This could have been remedied by validating key observations from the study using traditional methods such as flow cytometry and qPCR, but the limited number of validation experiments did not support the conclusions of the scRNA seq analysis. An important control group (PG:VG) had extremely low cell numbers and was basically not useful. Statistical analysis is lacking in almost all figures. Overall, this is a preliminary study with some potentially interesting observations, but no solid conclusions can be made from the data presented.The only new validation experiment is the immunofluorescent staining of neutrophils in Figure 4. The images are very low resolution and low quality and it is not clear which cells are neutrophils. S100A8 (calprotectin) is highly abundant in neutrophils but not strictly neutrophil-specific. It's hard to distinguish positive cells from autofluorescence in both Ly6g and S100a8 channels. No statistical analysis in the quantification.

We thank the reviewer for identifying the strengths of this study and pointing out the gaps in knowledge. Overall, our purpose to present this data is to provide the scRNA seq results as a resource to a wider community. We have used techniques like flow cytometry, multianalyte cytokine array and immunofluorescence to validate some of the results. We agree with the reviewer that we were unable to rightly point out the significance of our findings with the immunofluorescent stain in the previous edit. We have revised the manuscript and included the quantification for both Ly6G+ and S100A8+ cells in e-cig aerosol exposed and control lung tissues. Briefly, we identified a marked decrease in the staining for S100A8 (marker for neutrophil activation) in tobacco-flavored e-cig exposed mouse lungs as compared to controls. Upon considering the corroborating evidence from scRNA seq and flow cytometry with regards to increased neutrophil percentages in experimental group and lowered staining for active neutrophils using immunofluorescence, we speculate that exposure to e-cig (tobacco) aerosols may alter the neutrophil dynamics within the lungs. Also, co-immunofluorescence identified a more prominent co-localization of the two markers in control samples as compared to the treatment group which points towards some changes in the innate immune milieu within the lungs upon exposures. Future work is required to validate these speculations.

We have now discussed all the above-mentioned points in the Discussion section of the revised manuscript and toned down our conclusions regarding sex-dependent changes from scRNA seq data.

It is unclear what the meaning of Fig. 3A and B is, since these numbers only reflect the number of cells captured in the scRNAseq experiment and are not biologically meaningful. Flow cytometry quantification is presented as cell counts, but the percentage of cells from the CD45+ gate should be shown. No statistical analysis is shown, and flow cytometry results do not support the conclusions of scRNAseq data.

We thank the reviewer for this question. However, we would like to highlight that scRNA seq and flow cytometry may show similar trends but cannot be identical as one relies on cell surface markers (protein) for identification of cell types, while other is dependent on the transcriptomic signatures to identify the cell types. In our data, for the myeloid cells (alveolar macrophages and neutrophils), the scRNA and flow cytometry data match in trend. However, the trends do not match with respect to the lymphoid cells being studied (CD4 and CD8 T cells). The possible explanation for such a finding could be possible high gene dropout rates in scRNA seq, different analytical resolution for the two techniques and pooling of samples in our single cell workflow. We realize these shortcomings in our analyses and mention it clearly in the discussion as limitation of our work. It is important to note also that cell frequencies identified in scRNA seq just provide wide and indistinct indications which need to be further validated, which we tried to accomplish in our work to some degree. Our flow-based results clearly highlight the sex-specific variations in the immune cell percentages (something we could not have anticipated earlier). In future studies, we will include more replicates to tease out sex-based variations upon acute and chronic exposure to e-cig aerosols.

We have now replotted the graphs in Fig 3A and B and plotted the flow quantification as the percentage of total CD45+ cells. The gating strategy for the flow plots is also included as Figure S6 in the revised manuscript.

**Reviewer #2 (Public review):**
This study provides some interesting observations on how different flavour e-cigarettes can affect lung immunology; however, there are numerous flaws, including a low replicate number and a lack of effective validation methods, meaning findings may not be repeated. This is a revised article but several weaknesses remain related to the analysis and interpretation of the data.Strengths:The strength of the study is the successful scRNA-seq experiment which gives some preliminary data that can be used to create new hypotheses in this area.Weaknesses:Although some text weaknesses have been addressed since resubmission, other specific weaknesses remain: The major weakness is the n-number and analysis methods. Two biological n per group is not acceptable to base any solid conclusions. Any validatory data was too little (only cell % data) and not always supporting the findings (e.g. figure 3D does not match 3B/4A). Other examples include:There aren't enough cells to justify analysis - only 300-1500 myeloid cells per group with not many of these being neutrophils or the apparent 'Ly6G- neutrophils'.

We thank the reviewer for the comment, but we disagree with the reviewer in terms of the justification of analyses. All the flavored e-cig aerosol groups were compared with air controls to deduce the outcomes in the current study. We already acknowledge low sample quality for PGVG group and have only included the comparisons with PGVG upon reviewer’s request which is open to interpretation by the reader.

By that measure, each treatment group (except PGVG group) has over 1000 cells with 24777 genes being analyzed for each cell type, which by the standards of single cell is sufficient. We understand that this strategy should not be used for detection of rare cell populations, which was neither the purpose of this manuscript nor was attempted. We conduct comparisons of broader cell types and mention more samples need to be added in the Discussion section of the revised manuscript.

As for the Ly6G neutrophil category, we don’t only base our results on scRNA analyses but also perform co-immunofluorescence and multi-analyte analyses and use evidence from previous literature to back our outcome. To avoid over-stating our results we have revamped the whole manuscript and ensured to tone down our results with relation to the presence of Ly6G- neutrophils. We do understand that more work is required in the future, but our work clearly shows the shift in neutrophil dynamics upon exposure which should be reported, in our opinion.

The dynamic range of RNA measurement using scRNAseq is known to be limited - how do we know whether genes are not expressed or just didn't hit detection? This links into the Ly6G negative neutrophil comments, but in general the lack of gene expression in this kind of data should be viewed with caution, especially with a low n number and few cells. The data in the entire paper is not strong enough to base any solid conclusion - it is not just the RNA-sequencing data.

We acknowledge this to be a valid point and have revamped the manuscript and toned down our conclusions. However, such limitations exist with any scRNA seq dataset and so must be interpreted accordingly by the readers. We do understand that due to the low cell counts and the limitations with scRNA seq we should not perform DESeq2 analyses for Ly6G+ versus Ly6G- neutrophil categories, which was never attempted at the first place. However, our results with co-immunofluorescence, multianalyte assay and scRNA expression analyses in myeloid cluster do point towards a shift in neutrophil activation which needs to be further investigated. Furthermore, Ly6G deficiency has been linked to immature neutrophils in many previous studies and is not an unlikely outcome that needs to be treated with immense skepticism.

We wish to make this dataset available as a resource to influence future research. We are aware of its limitations and have been transparent with regards to our experimental design, capture strategy, the quality of obtained results, and possible caveats to make it is open for discussion by the readers.

There is no data supporting the presence of Ly6G negative neutrophils. In the flow cytometry only Ly6G+ cells are shown with no evidence of Ly6G negative neutrophils (assuming equal CD11b expression). There is no new data to support this claim since resubmission and the New figures 4C and D actually show there are no Ly6G negative cells - the cells that the authors deem Ly6G negative are actually positive - but the red overlay of S100A8 is so strong it blocks out the green signal - looking to the Ly6G single stains (green only) you can see that the reported S100A8+Ly6G- cells all have Ly6G (with different staining intensities).

We thank the reviewer for this query and do understand the skepticism. We have now quantified the data to provide more clarity for interpretation. As we were using paraffin embedded tissues, some autofluorescence is expected which could explain some of reviewer’s concerns. However we expect that the inclusion of better quality images and quantification must address some of the concerns raised by the reviewer.

Eosinophils are heavily involved in lung macrophage biology, but are missing from the analysis - it is highly likely the RNA-sequence picked out eosinophils as Ly6G- neutrophils rather than 'digestion issues' the authors claim

We thank the reviewer for raising a valid concern. However, the Ly6G- cluster cannot be eosinophils in our case. Literature suggests SiglecF as an important biomarker of eosinophils which was absent in the Ly6G- cluster our in scRNA seq analyses as shown in File S18 and Figure 6B of the revised manuscript. We have now provided a detailed explanation (Lines 476-488; 503-506) of the observed results pertaining to eosinophil population in the revised manuscript to further address some of the concerns raised by this reviewer.

After author comments, it appears the schematic in Figure 1A is misleading and there are not n=2/group/sex but actually only n=1/group/sex (as shown in Figure 6A). Meaning the n number is even lower than the previous assumption.

We concur with reviewers’ valid concern and so are willing to provide this data as a resource for a wider audience to assist future work. Pooling of samples have been practiced by many groups previously to save resources and expense. We did it for the very same reason. It may not be the preferred approach, but it still has its merit considering the vast amount of cell-specific data generated using this strategy. To avoid overstating our results we have ensured to maintain transparency in our reporting and acknowledge all the limitations of this study.

We do not believe that the strength of scRNA seq lies in drawing conclusive results, but to tease our possible targets and direction that need to be validated with more work. In that respect, our study does identify the target cell types and biological processes which could be of importance for future studies.

**Reviewer #3 (Public review):**
This work aims to establish cell-type specific changes in gene expression upon exposure to different flavors of commercial e-cigarette aerosols compared to control or vehicle. Kaur et al. conclude that immune cells are most affected, with the greatest dysregulation found in myeloid cells exposed to tobacco-flavored e-cigs and lymphoid cells exposed to fruit-flavored e-cigs. The up- and down-regulated genes are heavily associated with innate immune response. The authors suggest that a Ly6G-deficient subset of neutrophils is found to be increased in abundance for the treatment groups, while gene expression remains consistent, which could indicate impaired function. Increased expression of CD4+ and CD8+ T cells along with their associated markers for proliferation and cytotoxicity is thought to be a result of activation following this decline in neutrophil-mediated immune response.Strengths:Single cell sequencing data can be very valuable in identifying potential health risks and clinical pathologies of lung conditions associated with e-cigarettes considering they are still relatively new.Not many studies have been performed on cell-type specific differential gene expression following exposure to e-cig aerosols.The assays performed address several factors of e-cig exposure such as metal concentration in the liquid and condensate, coil composition, cotinine/nicotine levels in serum and the product itself, cell types affected, which genes are up- or down-regulated and what pathways they control.Considerations were made to ensure clinical relevance such as selecting mice whose ages corresponded with human adolescents so that data collected was relevant.Weaknesses:The exposure period of 1 hour a day for 5 days is not representative of chronic use and this time point may be too short to see a full response in all cell types. The experimental design is not well-supported based on the literature available for similar mouse models. Clinical relevance of this short exposure remains unclear.

We thank the reviewer for this query. However, we would like to emphasize that chronic exposure was never the intention of this study. We wished to design a study for acute nose-only exposure owing to which the study duration was left shorter. Shorter durations limit the stress and discomfort to the animal. The in vivo study using nose-only exposure regimen is still developing with multiple exposure regimen being used by different groups. To our knowledge there is no gold standard of e-cig aerosol exposure which is widely accepted other than the CORESTA recommendations, which we followed. Also, we show in our study how the daily exposure to leached metals vary in a flavor-dependent manner thus validating that exposure regime does need more attention in terms of equal dosing, particle distribution and composition- something we have started doing in our future studies. We have included all the explanations in the revised manuscript (Lines 82-85, 425-435, 648-654).

Several claims lack supporting evidence or use data that is not statistically significant. In particular, there were no statistical analyses to compare results across sex, so conclusions stating there is a sex bias for things like Ly6G+ neutrophil percentage by condition are observational.

We agree with reviewer’s comment and have taken this into consideration. We have now revamped the whole manuscript and toned down most of the sex-based conclusions stated in this work. Having said that, it is important to note that most of the work relying solely on scRNA seq, as is the case for this study, is observational in nature and needs to be assessed bearing this in mind.

Overall, the paper and its discussion are relatively surface-level and do not delve into the significance of the findings or how they fit into the bigger picture of the field. It is not clear whether this paper is intended to be used as a resource for other researchers or as an original research article.

We have now reworked on the Discussion and tried to incorporate more in-depth discussion and the results providing our insights regarding the observations, discrepancies and the possible explanations. We have also made it clear that this paper is intended to be used as a resource by other researchers (Lines 577-579)

The manuscript has some validation of findings but not very comprehensive.

We have now revamped the manuscript. We have Included quantification for immunofluorescence data with better representation of the GO analyses. We have worked on the Results and Discussion sections to make this a useful resource for the scientific community.

This paper provides a strong foundation for follow-up experiments that take a closer look at the effects of e-cig exposure on innate immunity. There is still room to elaborate on the differential gene expression within and between various cell types.

We thank the reviewer for pointing out the strength of this paper. The reason why we refrained from elaborating of the differential gene expressions within and between various cell types was due to low sample number and sequencing depth for this study. However the raw data will be provided with the final publication, which should be freely accessible to the public to re-analyze the data set as they deem fit.

Comments on revisions:The reviewers have addressed major concerns with better validation of data and improved organization of the paper. However, we still have some concerns and suggestions pertaining to the statistical analyses and justifications for experimental design.We appreciate the nuance of this experimental design, and the reviewers have adequately commented on why they chose nose-only exposure over whole body exposure. However, the justification for the duration of the exposure, and the clinical relevance of a short exposure, have not been addressed in the revised manuscript.

We thank the editor for this query. We have now addressed this query briefly in Lines 82-85, 425-435, 648-654 of the revised manuscript. We would like to add, however, that we intend to design a study for acute nose-only exposure for this project. Shorter durations limit the stress and discomfort to the animal, owing to which a duration of 1hour per day was chosen. The in vivo study using nose-only exposure regimen is still developing with multiple exposure regimen being used by different groups. Ours is one such study in that direction just intended to identify cell-specific changes upon exposure. Considering our results in Figure 1B showing variations in the level of metals leached in each flavor per day, the appropriate exposure regimen to design a controlled, reproducible experiment needs to be discussed. There could be room for improvement in our strategy, but this was the best regimen that we found to be appropriate per the literature and our prior knowledge in the field.

The presentation of cell counts should be represented by a percentage/proportion rather than a raw number of cells. Without normalization to the total number of cells, comparisons cannot be made across groups/conditions. This comment applies to several figures.

We thank the editor for this comment and have now made the requested change in the revised manuscript.

We appreciate that the authors have taken the reviewers' advice to validate their findings. However, we have concerns regarding the immunofluorescent staining shown in Figure 4. If the red channel is showing a pan-neutrophil marker (S100A8) and the green channel is showing only a subset of neutrophils (LY6G+), then the green channel should have far less signal than the red channel. This expected pattern is not what is shown in the figure, with the Ly6G marker apparently showing more expression than S100A8. Additionally, the FACS data states that only 4-5% of cells are neutrophils, but the red channel co-localizes with far more than 4-5% of the DAPI stain, meaning this population is overrepresented, potentially due to background fluorescence (noise). In addition, some of the shapes in the staining pattern do not look like true neutrophils, although it is difficult to tell because there remains a lot of background staining. The authors need to verify that their S100A8 and Ly6G antibodies work and are specific to the populations they intend to target. It is possible that only the brightest spots are truly S100A8+ or Ly6G+.

We thank the editor for this comment and acknowledge that we may have made broad generalizations in our interpretation of our data previously. We have now revisited the data and quantified the two fluorescence for better interpretation of our results. We have also reassessed our conclusions from this data and reworded the manuscript accordingly. Briefly we believe that Ly6G deficiency could be an indication of the presence of immature neutrophils in the lungs. This is a common process of neutrophil maturation. An active neutrophil population has Ly6G and should also express S100A8 indicating a normal neutrophilic response against stressors. However, our results, despite some autofluorescence which is common with lung tissues, shows a marked decline in the S100A8+ cells in the lung of tobacco-flavored e-cig aerosol exposed mice as compared to air controls. We also do not see prominent co-localization of the two markers in exposed group thus proving a shift in neutrophil dynamics which requires further investigation. We would also like to mention here that S100A8 is predominantly expressed in neutrophils, but is also expressed by monocytes and macrophages, so that could explain the over-representation of these cells in our immunofluorescence results. We have now included this in the Discussion section (Lines 489- 538) of the revised manuscript.

Paraffin sections do not always yield the best immunostaining results and the images themselves are low magnification and low resolution.

We agree with the editor that paraffin sections may not yield best results, we have worked on the final figure to improve the quality of the displayed results and zoomed-in some parts of the merged image to show the differences in the co-localization patterns for the two markers in our treated and control groups for easier interpretation.

Please change the scale bars to white so they are more visible in each channel.

The merged image in Figure 6C now has a white scale bar.

We appreciate that this is a preliminary test used as a resource for the community, but there is interesting biology regarding immune cells that warrants DEG analysis by the authors. This computational analysis can be easily added with no additional experiments required.

We thank the editor for this comment and agree that interesting biology regarding immune cells could be explored upon performing the DEG analyses on individual immune populations. However, due to the small sample size, low sequencing depth and pooling of same sex animals in each treatment group, we refrained from performing that analyses fearing over-representation of our results. We will be providing the link to the raw data with this publication which will be freely accessible to public on NIH GEO resource to allow further analyses on this dataset by the judgement of the investigator who utilizes it as a resource.

**Recommendations for the authors:**

**Reviewer #1 (Recommendations for the authors):**
(Minor) The pathway analyses in Fig. 6-8 have different fonts than what's used in all other figures.

We have now made the requested change in the revised manuscript.